# Unsupervised Parallel MRI Reconstruction via Projected Conditional Flow Matching

## Abstract

Reconstructing high-quality images from substantially undersampled k-space data for accelerated MRI presents a challenging ill-posed inverse problem. Supervised deep learning has transformed the field by using large amounts of fully sampled ground-truth MR images, either to directly reconstruct undersampled data into fully sampled images with neural networks, or to learn the prior distribution of fully sampled images through generative models. However, in practical scenarios, acquiring ground-truth fully sampled MRI images is not viable due to the inherently slow nature of its data acquisition process. Despite advances in self-supervised/unsupervised MRI reconstruction, the performance remains inadequate at high acceleration rates. To address these gaps, we introduce the Projected Conditional Flow Matching (PCFM) and its unsupervised transformation, which is designed to learn the prior distribution of fully sampled parallel MRI by solely utilizing the undersampled k-space measurements. To reconstruct the image, we establish a novel relationship between the marginal vector field in the measurement space, which generates the associated probability flow in terms of the continuity equation, and the optimal solution to PCFM. This connection results in a cyclic dual-space sampling algorithm for unsupervised reconstruction. Our method was evaluated against contemporary state-of-the-art supervised, self-supervised, and unsupervised baseline techniques on parallel MRI using publicly available datasets fastMRI and CMRxRecon. Experimental results show that our technique significantly surpasses existing self-supervised and unsupervised baselines, while also yielding better performance than most supervised methods. Our code will be available at https://github.com/anonymous.

## 1 Introduction

Magnetic Resonance Imaging (MRI) is a cornerstone of modern medical diagnostics, providing exceptional soft-tissue contrast without the use of ionizing radiation. However, a significant clinical limitation of MRI is its inherently slow data acquisition speed. The time required for a scan is directly proportional to the amount of data that must be acquired in the MRI raw data space, known as k-space. The development of multi-coil receiver arrays, replacing single coils with smaller, localized ones (Roemer et al., 1990; Sodickson & Manning, 1997), enabled accelerated data acquisition through parallel imaging. Key algorithms, SENSitivity Encoding (SENSE) (Pruessmann et al., 1999) and GeneRalized Autocalibrating Partially Parallel Acquisitions (GRAPPA) (Griswold et al., 2002), laid the groundwork for this. By combining parallel imaging with sparsity-promoting terms, later compressed sensing (CS)-based methods achieve higher acceleration than traditional techniques (Lustig et al., 2007; 2008).

Building on this concept, data-driven models, particularly physics-informed "unrolled" neural networks that emulate classical iterative optimization methods and leverage prior knowledge encoded by convolutional neural networks (CNNs) (Ulyanov et al., 2018), attain cutting-edge outcomes in terms of both reconstruction quality and speed (Aggarwal et al., 2018; Hammernik et al., 2018). More recently, advancements in accelerated MRI reconstruction have been pursued through modern generative models. These generative models, rather than focusing on learning a single point estimate, are designed to approximate the entire probability distribution of high-quality MR images (Mardani et al., 2018; Tezcan et al., 2018; Song et al., 2022; Chung & Ye, 2022). This approach aids in solving the inverse problem by enabling sampling from the posterior distribution $p(\boldsymbol{x} \mid \boldsymbol{y})$,

where $\boldsymbol{x}$ represents the target fully sampled image, and $\boldsymbol{y}$ includes the undersampled multi-coil k-space measurement. Despite achieving high reconstruction accuracy, these supervised frameworks necessitate access to vast collections of fully sampled ground-truth images for training, which are not only costly but also commonly unattainable. Emerging self-supervised techniques aim to reduce the reliance on fully sampled MRI datasets during training (Wang et al., 2025). Nevertheless, they fall short in accuracy when dealing with highly undersampled data, e.g., $8\times$ accelerated MRI.

To address these challenges, we introduce an unsupervised generative model for parallel MRI reconstruction requiring solely undersampled k-space data for training. Drawing inspiration from the generalized Stein's Unbiased Risk Estimator (Stein, 1981; Eldar, 2008), we present the projected conditional flow matching (PCFM) objective alongside its unsupervised adaptation, which facilitates learning the prior distribution of fully sampled MRI using only undersampled data. Given that a closed-form solution to the projection operator is intractable for parallel MRI, we propose to employ a numerical method to approximate the projection operator during training. Subsequently, we derive a new connection between the probability flow in the measurement space under projection and the optimal solution to the proposed PCFM objective, introducing a reconstruction algorithm based on the optimal PCFM solution. The proposed approach is also capable of dealing with noisy measurements under the formulation. Our framework is evaluated using two public parallel brain and cardiac MRI datasets, demonstrating superior reconstruction performance over existing baselines, even when trained solely with undersampled k-space measurements.

## 2 BACKGROUND

### 2.1 PARALLEL MRI RECONSTRUCTION

The fundamental principle of parallel MRI builds on the fact that each coil in an array receives a distinct spatial sensitivity profile, that is, a unique spatially weighted view of the underlying anatomy. This spatial encoding ability can be used to compensate for the spatial information lost when k-space is undersampled. Formally, the forward model of parallel (also known as multi-coil) MRI writes

$$\boldsymbol{y}_s = \boldsymbol{A}_s \boldsymbol{x} + \boldsymbol{e}, \tag{1}$$

where $s$ indexes the randomness in the undersampling mask, $\boldsymbol{x} \in \mathcal{X} \subset \mathbb{C}^D$ is the underlying fully sampled complex-valued image, $\boldsymbol{y}_s = [\boldsymbol{y}_{s,1}^\mathsf{T}, \ldots, \boldsymbol{y}_{s,C}^\mathsf{T}]^\mathsf{T} \in \mathcal{Y} \subset \mathbb{C}^{Cd}$ is the acquired k-space measurements from $C$ receiver coils with $d \leq D$, and $\boldsymbol{e} \sim \mathcal{CN}(\boldsymbol{0}, \sigma_0^2 \boldsymbol{I}_{Cd})$ denotes measurement noise. In particular, the *coil-combined* forward operator $\boldsymbol{A}_s$ is defined as

$$\boldsymbol{A}_s \triangleq \begin{bmatrix} \boldsymbol{M}_s \boldsymbol{F} \boldsymbol{S}_1 \\ \vdots \\ \boldsymbol{M}_s \boldsymbol{F} \boldsymbol{S}_C \end{bmatrix} \in \mathbb{C}^{Cd \times D}, \tag{2}$$

which is composed of the undersampling mask $\boldsymbol{M}_s \in \{0,1\}^{d \times D}$, the discrete Fourier transform $\boldsymbol{F} \in \mathbb{C}^{D \times D}$, and diagonal matrices $\boldsymbol{S}_c \in \mathbb{C}^{D \times D}$ representing the sensitivity maps. For parallel MRI with acceleration factor $\alpha \triangleq D/d > 1$, Eq. (2) can be rank-deficient with a non-zero null space, leading to a challenging ill-posed inverse problem. For simplicity, we assume that the randomness in $s$ is incorporated in $\boldsymbol{y}$ in the following, and thus denote the forward operator as $\boldsymbol{A}$.

### 2.2 CONDITIONAL FLOW MATCHING

Conditional flow matching (CFM) (Lipman et al., 2023) provides a simulation-free technique to learn a continuous normalizing flow (Chen et al., 2018) that transforms a base distribution $p_1$ to a target distribution $p_0$. In this work, we assume $p_1^X = \mathcal{CN}(\boldsymbol{0}, 2\boldsymbol{I}_D)$, and $p_0^X$ produces the underlying fully sampled images, where the superscript $X$ indicates that the flow is in the fully sampled image space $\mathcal{X}$. This transformation can be specified by an ordinary differential equation (ODE) with a time-dependent smooth vector field $\boldsymbol{u}_t^X : [0,1] \times \mathbb{C}^D \to \mathbb{C}^D$, i.e., $\mathrm{d}\boldsymbol{x}_t = \boldsymbol{u}_t^X(\boldsymbol{x}_t)\mathrm{d}t$. This induces a probability path $p_t^X$ as the push-forward distribution of $p_0^X$ by the ODE dynamics, satisfying the continuity equation (Villani et al., 2009):

$$\frac{\partial p_t^X}{\partial t} + \nabla \cdot (p_t^X \boldsymbol{u}_t^X) = 0. \tag{3}$$

CFM proposes to construct such a marginal vector field $\boldsymbol{u}_t^X$ by introducing the conditional variable $\boldsymbol{z}^X \sim q(\boldsymbol{z}^X)$ and conditional vector field $\boldsymbol{u}_t^X(\boldsymbol{x} \mid \boldsymbol{z}^X)$. In this paper, we consider the popular choice of $\boldsymbol{z}^X = (\boldsymbol{x}_0, \boldsymbol{x}_1)$ and the independent coupling $q = p_0^X \times p_1^X$ as generalized by (Tong et al., 2023). Then, we assume a conditional probability path $p_t^X(\cdot \mid \boldsymbol{z}^X)$ that satisfies the boundary conditions $p_0^X = \mathbb{E}_{q(\boldsymbol{z}^X)} \left[ p_0^X(\cdot \mid \boldsymbol{z}^X) \right]$ and $p_1^X = \mathbb{E}_{q(\boldsymbol{z}^X)} \left[ p_1^X(\cdot \mid \boldsymbol{z}^X) \right]$. One example is the conditional optimal transport (OT) path $p_t^X(\boldsymbol{x} \mid \boldsymbol{z}^X) = \delta_{a_t \boldsymbol{x}_0 + b_t \boldsymbol{x}_1}(\boldsymbol{x})$ with $a_t = 1 - t$ and $b_t = t$ (Lipman et al., 2023; Liu et al., 2023). This leads to the conditional vector field $\boldsymbol{u}_t^X(\boldsymbol{x} \mid \boldsymbol{z}^X) = a_t' \boldsymbol{x}_0 + b_t' \boldsymbol{x}_1$ by the continuity equation w.r.t. $p_t^X(\boldsymbol{x} \mid \boldsymbol{z}^X)$, where $a_t' \triangleq \frac{\mathrm{d}a_t}{\mathrm{d}t}$ and $b_t' \triangleq \frac{\mathrm{d}b_t}{\mathrm{d}t}$. Then by verifying Eq. (3), we can show that the marginal vector field

$$\boldsymbol{u}_t^X(\boldsymbol{x}) \triangleq \mathbb{E}_{q(\boldsymbol{z}^X)} \left[ \boldsymbol{u}_t^X(\boldsymbol{x} \mid \boldsymbol{z}^X) \frac{p_t^X(\boldsymbol{x} \mid \boldsymbol{z}^X)}{p_t^X(\boldsymbol{x})} \right] \tag{4}$$

generates the probability flow $p_t^X$. Meanwhile, learning of the flow is achieved by minimizing the conditional flow matching objective:

$$\mathcal{L}_{\mathrm{CFM}}(\boldsymbol{\theta}) \triangleq \mathbb{E}_{t, q(\boldsymbol{z}^X), p_t^X(\boldsymbol{x} \mid \boldsymbol{z}^X)} \left\| \boldsymbol{h}_{\boldsymbol{\theta}}^X(\boldsymbol{x}, t) - \boldsymbol{u}_t^X(\boldsymbol{x} \mid \boldsymbol{z}^X) \right\|_2^2, \tag{5}$$

where $\boldsymbol{h}_{\boldsymbol{\theta}}^X(\cdot, t)$ is a network that predicts the $\mathcal{X}$-space marginal vector field.

## 3 METHOD

This section introduces the proposed framework designed to reconstruct parallel MRI using only undersampled k-space data. The framework integrates two principal components: (1) the projected conditional flow matching (PCFM) objective along with its unsupervised transformation as a new formulation for learning the prior (Section 3.1), and (2) a new reconstruction algorithm for inference that exploits the relationship between measurement-space probability flow and the optimal solution to PCFM (Section 3.2).

### 3.1 PROJECTED CONDITIONAL FLOW MATCHING

Due to the scarcity of fully sampled ground-truth signal $\boldsymbol{x}_0$, optimizing the $\mathcal{X}$-space CFM objective from Eq. (5) using a large dataset of fully sampled MRI scans is impractical because the conditional path and vector field within the $\mathcal{X}$-space are not accessible. To address this, we introduce a $\mathcal{Y}$-space conditional probability path $p_t^Y(\boldsymbol{y} \mid \boldsymbol{z}^Y) = \delta_{a_t \boldsymbol{y}_0 + b_t \boldsymbol{y}_1}(\boldsymbol{y})$, where $\boldsymbol{z}^Y \triangleq (\boldsymbol{y}_0, \boldsymbol{y}_1)$, in which $\boldsymbol{y}_0 = \boldsymbol{A}\boldsymbol{x}_0 + \boldsymbol{e}_0$ is the undersampled k-space measurements from parallel MRI, and $\boldsymbol{y}_1 = \boldsymbol{A}\boldsymbol{x}_1 + \boldsymbol{e}_1$ with $\boldsymbol{e}_1 \sim \mathcal{CN}(\boldsymbol{0}, \sigma_1^2 \boldsymbol{I}_{Cd})$ is the projected noise sampled from the base distribution $p_1^X$. This leads to

$$p_t^Y(\boldsymbol{y} \mid \boldsymbol{z}^X) = \int p_t^Y(\boldsymbol{y} \mid \boldsymbol{z}^Y) p(\boldsymbol{z}^Y \mid \boldsymbol{z}^X) \mathrm{d}\boldsymbol{z}^Y$$
$$= \mathcal{CN}\left( \boldsymbol{y} \mid \boldsymbol{A}(a_t \boldsymbol{x}_0 + b_t \boldsymbol{x}_1), (a_t^2 \sigma_0^2 + b_t^2 \sigma_1^2) \boldsymbol{I}_{Cd} \right). \tag{6}$$

Fig. 1 depicts the graphical model of the random variables.

Since $\boldsymbol{x}_0$ and $\boldsymbol{u}_t^X(\boldsymbol{x} \mid \boldsymbol{z}^X) = a_t' \boldsymbol{x}_0 + b_t' \boldsymbol{x}_1$ are unknown, and the forward operator $\boldsymbol{A}$ is rank-deficient, we can only expect to optimize the CFM objective Eq. (5) in the range space $\mathcal{R}(\boldsymbol{A}^\top)$ of $\boldsymbol{A}^\top$ (Eldar, 2008). Therefore, we propose to optimize the following objective function

$$\mathcal{L}_{\mathrm{PCFM}}(\boldsymbol{\theta}) \triangleq \mathbb{E}_{t, q(\boldsymbol{z}^X), p_t^X(\boldsymbol{x} \mid \boldsymbol{z}^X), p_t^Y(\boldsymbol{y} \mid \boldsymbol{z}^X)} \left\| \boldsymbol{P} \left[ \boldsymbol{v}_{\boldsymbol{\theta}}^X(\boldsymbol{y}, t) - \boldsymbol{u}_t^X(\boldsymbol{x} \mid \boldsymbol{z}^X) \right] \right\|_2^2, \tag{7}$$

where $\boldsymbol{P} \triangleq \boldsymbol{A}^+ \boldsymbol{A}$ is the orthogonal projection onto the range space $\mathcal{R}(\boldsymbol{A}^\top)$, with $\boldsymbol{A}^+$ denoting the Moore-Penrose pseudoinverse of $\boldsymbol{A}$. Intuitively, this objective projects the CFM error onto the

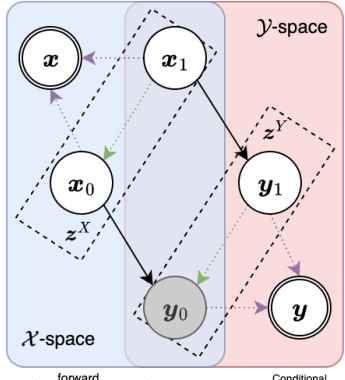

Figure 1: Generation chart of the dual-space conditional probability paths, where observed variables are shaded, and deterministic variables are in double circles. Dotted green arrows indicate deterministic ODE flows, whereas purple ones denote conditional OT paths. The path from $\boldsymbol{x}_1$ to $\boldsymbol{y}_0$ is traversed either through the $\mathcal{X}$-space OR the $\mathcal{Y}$-space diagram, as indicated by the background in different colors.

subspace "visible" to the k-space measurements from the mask $\boldsymbol{M}$. Thus, we dub this objective function *projected conditional flow matching* (PCFM).

We note that within the framework of the parallel MRI forward model, a closed-form solution for this projection is not available. To tackle this issue, we note the relationships $\boldsymbol{P} = \boldsymbol{A}^+\boldsymbol{A} = (\boldsymbol{A}^*\boldsymbol{A})^+\boldsymbol{A}^*\boldsymbol{A}$ and recognize that $\boldsymbol{A}^*\boldsymbol{A}$ is a positive semi-definite (PSD) matrix. Consequently, we utilize a numerical approximation through the conjugate gradient (CG) method (Appendix C.1), which is suitable for solving the linear system $\boldsymbol{A}^*\boldsymbol{A}\boldsymbol{r}_{\boldsymbol{\theta}}^X = \boldsymbol{A}^*\boldsymbol{A}\left[\boldsymbol{v}_{\boldsymbol{\theta}}^X(\boldsymbol{y},t) - \boldsymbol{u}_t^X(\boldsymbol{x} \mid \boldsymbol{z}^X)\right]$, where $\boldsymbol{r}_{\boldsymbol{\theta}}^X = \boldsymbol{P}\left[\boldsymbol{v}_{\boldsymbol{\theta}}^X(\boldsymbol{y},t) - \boldsymbol{u}_t^X(\boldsymbol{x} \mid \boldsymbol{z}^X)\right]$. The optimal solution to the PCFM objective is formalized by the following proposition. The proofs can be found in Appendix A.

**Proposition 1 (Optimal solution to PCFM).** *The minimizer of the PCFM objective is given by*

$$\boldsymbol{v}_{\boldsymbol{\theta}^*}^X(\boldsymbol{y},t) = \mathbb{E}_{q_t(\boldsymbol{z}^X|\boldsymbol{y}),p_t^X(\boldsymbol{x}|\boldsymbol{z}^X)}\left[\boldsymbol{u}_t^X(\boldsymbol{x} \mid \boldsymbol{z}^X)\right] + \boldsymbol{w}, \tag{8}$$

*where $q_t(\boldsymbol{z}^X \mid \boldsymbol{y}) = \frac{q(\boldsymbol{z}^X)p_t^Y(\boldsymbol{y}|\boldsymbol{z}^X)}{p_t^Y(\boldsymbol{y})}$, and $\boldsymbol{w}$ is any vector in the null space of $\boldsymbol{A}$, i.e., $\boldsymbol{A}\boldsymbol{w} = \boldsymbol{P}\boldsymbol{w} = \boldsymbol{0}$. In particular, when $\boldsymbol{u}_t^X(\boldsymbol{x} \mid \boldsymbol{z}^X) = a_t'\boldsymbol{x}_0 + b_t'\boldsymbol{x}_1$ that is independent of $\boldsymbol{x}$, we have*

$$\boldsymbol{v}_{\boldsymbol{\theta}^*}^X(\boldsymbol{y},t) = \mathbb{E}_{q_t(\boldsymbol{z}^X|\boldsymbol{y})}\left[\boldsymbol{u}_t^X(\boldsymbol{x} \mid \boldsymbol{z}^X)\right] + \boldsymbol{w}. \tag{9}$$

However, the PCFM objective Eq. (7) still depends on the unknown fully sampled MRI $\boldsymbol{x}_0$ through Monte Carlo sampling from $q(\boldsymbol{z}^X)$ during training. To address this, inspired by the generalized Stein's unbiased estimator (Eldar, 2008), we propose to construct an unbiased estimate of the PCFM objective that does not involve the unknown $\boldsymbol{x}_0$. To this end, we notice an induced linear forward model between the dual-space conditional vector fields

$$\boldsymbol{u}_t^Y(\boldsymbol{y} \mid \boldsymbol{z}^Y) = a_t'\boldsymbol{y}_0 + b_t'\boldsymbol{y}_1 = \boldsymbol{A}\boldsymbol{u}_t^X(\boldsymbol{x} \mid \boldsymbol{z}^X) + a_t'\boldsymbol{e}_0 + b_t'\boldsymbol{e}_1, \tag{10}$$

and the deterministic mapping between the conditional path and the conditional vector field

$$\boldsymbol{y} = \frac{a_t}{a_t'}\boldsymbol{u}_t^Y(\boldsymbol{y} \mid \boldsymbol{z}^Y) - b_t'\left(\frac{a_t}{a_t'} - \frac{b_t}{b_t'}\right)\boldsymbol{y}_1. \tag{11}$$

Based on this, we derive the following unsupervised transformation of the PCFM objective, which does not require fully sampled MRI data for training the vector field predictor.

**Proposition 2 (Unsupervised transformation of PCFM).** *Assuming deterministic conditional probability paths $\boldsymbol{x} = a_t\boldsymbol{x}_0 + b_t\boldsymbol{x}_1$ and $\boldsymbol{y} = a_t\boldsymbol{y}_0 + b_t\boldsymbol{y}_1$ with $\boldsymbol{y}_0 = \boldsymbol{A}\boldsymbol{x}_0 + \boldsymbol{e}_0$ and $\boldsymbol{y}_1 = \boldsymbol{A}\boldsymbol{x}_1 + \boldsymbol{e}_1$, where $\boldsymbol{e}_0 \sim \mathcal{CN}(\boldsymbol{0}, \sigma_0^2\boldsymbol{I}_{Cd})$ and $\boldsymbol{e}_1 \sim \mathcal{CN}(\boldsymbol{0}, \sigma_1^2\boldsymbol{I}_{Cd})$, then up to a constant the PCFM objective can be transformed to*

$$\mathbb{E}_{t,q(\boldsymbol{z}^Y),p_t^Y(\boldsymbol{y}|\boldsymbol{z}^Y)}\left[\left\|\boldsymbol{P}\left[\boldsymbol{v}_{\boldsymbol{\theta}}^X(\boldsymbol{A}^*\boldsymbol{y},t) - \widehat{\boldsymbol{u}}_{t,ML}^X\right]\right\|_2^2 + \frac{2a_t}{a_t'}[(a_t'\sigma_0)^2 + (b_t'\sigma_1)^2]\nabla_{\boldsymbol{A}^*\boldsymbol{y}} \cdot \boldsymbol{P}\boldsymbol{v}_{\boldsymbol{\theta}}^X(\boldsymbol{A}^*\boldsymbol{y},t)\right], \tag{12}$$

*where $q(\boldsymbol{z}^Y) = q(\boldsymbol{y}_0)q(\boldsymbol{y}_1) = q(\boldsymbol{y}_0)\mathbb{E}_{q(\boldsymbol{x}_1)}[p(\boldsymbol{y}_1 \mid \boldsymbol{x}_1)]$ is sampled by the MRI forward model and Monte Carlo estimation, $\boldsymbol{P} = \boldsymbol{A}^+\boldsymbol{A}$ is the range-space projection, and*

$$\widehat{\boldsymbol{u}}_{t,ML}^X \triangleq (\boldsymbol{A}^*\boldsymbol{C}_t^{-1}\boldsymbol{A})^+\boldsymbol{A}^*\boldsymbol{C}_t^{-1}\boldsymbol{u}_t^Y(\boldsymbol{y} \mid \boldsymbol{z}^Y) \tag{13}$$

*with $\boldsymbol{C}_t = [(a_t'\sigma_0)^2 + (b_t'\sigma_1)^2]\boldsymbol{I}_d$ is the maximum likelihood solution of the forward model in Eq. (10). Note that $\boldsymbol{A}^+$ denotes the Moore-Penrose pseudoinverse of $\boldsymbol{A}$, which can be approximated by the conjugate gradient method (Appendix C.1).*

The network $\boldsymbol{v}_{\boldsymbol{\theta}}^X(\cdot,t)$ takes $\boldsymbol{A}^*\boldsymbol{y}$ as input to match the desired dimensionality of the architecture. By optimizing Eq. (12), the optimal solution to PCFM can be determined in an unsupervised learning fashion. The objective can also handle noisy measurements for $\sigma_0 > 0$. In the following section, we will delve into reconstructing a fully sampled image utilizing the obtained optimal PCFM solution.

## 3.2 RECONSTRUCTION VIA DUAL-SPACE CYCLIC FLOW INTEGRATION

In the inference phase, to reconstruct the fully sampled image $\boldsymbol{x}_0$ given the undersampled measure-memt $\boldsymbol{y}_0$, we note that given the generative model in Fig. 1, the posterior distribution $p(\boldsymbol{x}_0 \mid \boldsymbol{y}_0)$ can

be written as

$$p(\boldsymbol{x}_0 \mid \boldsymbol{y}_0) = \int p(\boldsymbol{x}_0 \mid \boldsymbol{x}_1, \boldsymbol{y}_1, \boldsymbol{y}_0) p(\boldsymbol{x}_1 \mid \boldsymbol{y}_1, \boldsymbol{y}_0) p(\boldsymbol{y}_1 \mid \boldsymbol{y}_0) \mathrm{d}\boldsymbol{x}_1 \mathrm{d}\boldsymbol{y}_1$$

$$= \int p(\boldsymbol{x}_0 \mid \boldsymbol{x}_1) p(\boldsymbol{x}_1 \mid \boldsymbol{y}_1) p(\boldsymbol{y}_1 \mid \boldsymbol{y}_0) \mathrm{d}\boldsymbol{x}_1 \mathrm{d}\boldsymbol{y}_1, \tag{14}$$

which can be evaluated by ancestral Monte-Carlo sampling as shown in Fig. 2. We detail the sampling procedure for each conditional distribution in the following paragraphs.

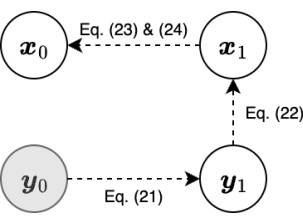

**Probability vector fields under projection and forward integration.** To sample from the distribution $p(\boldsymbol{y}_1 \mid \boldsymbol{y}_0)$, we note that given the conditional probability paths in dual spaces illustrated in Fig. 1, it is feasible to derive a specific relationship between the marginal vector field in the $\mathcal{Y}$-space and the optimal solution to the PCFM as discussed in Proposition 1, leading to a delta distribution of $p(\boldsymbol{y}_1 \mid \boldsymbol{y}_0)$ by the $\mathcal{Y}$-space ODE. Recall that the $\mathcal{Y}$-space conditional probability path is given by Eq. (6). Denote $\boldsymbol{\mu}_t(\boldsymbol{z}^X) \triangleq \boldsymbol{A}(a_t \boldsymbol{x}_0 + b_t \boldsymbol{x}_1)$ and $\sigma_t^2 = a_t^2 \sigma_0^2 + b_t^2 \sigma_1^2$. Then the time derivative of $p_t^Y(\boldsymbol{y} \mid \boldsymbol{z}^X)$ can be written as

Figure 2: Inference steps of the proposed reconstruction algorithm.

$$\frac{\partial p_t^Y(\boldsymbol{y} \mid \boldsymbol{z}^X)}{\partial t} = \frac{\partial p_t^Y(\boldsymbol{y} \mid \boldsymbol{z}^X)}{\partial \boldsymbol{\mu}_t} \cdot \frac{\mathrm{d}\boldsymbol{\mu}_t}{\mathrm{d}t} + \frac{\partial p_t^Y(\boldsymbol{y} \mid \boldsymbol{z}^X)}{\partial \sigma_t^2} \frac{\mathrm{d}\sigma_t^2}{\mathrm{d}t}$$

$$= -\nabla_{\boldsymbol{y}} p_t^Y(\boldsymbol{y} \mid \boldsymbol{z}^X) \cdot \boldsymbol{A}(a_t' \boldsymbol{x}_0 + b_t' \boldsymbol{x}_1) + \frac{1}{2} \Delta_{\boldsymbol{y}} p_t^Y(\boldsymbol{y} \mid \boldsymbol{z}^X)(2 a_t a_t' \sigma_0^2 + 2 b_t b_t' \sigma_1^2)$$

$$= -\nabla_{\boldsymbol{y}} \cdot \left( p_t^Y(\boldsymbol{y} \mid \boldsymbol{z}^X) \left[ \boldsymbol{A} \boldsymbol{u}_t^X(\boldsymbol{x} \mid \boldsymbol{z}^X) - (a_t a_t' \sigma_0^2 + b_t b_t' \sigma_1^2) \nabla_{\boldsymbol{y}} \log p_t^Y(\boldsymbol{y} \mid \boldsymbol{z}^X) \right] \right). \tag{15}$$

Therefore, by the continuity equation, we know that the $\mathcal{Y}$-space conditional vector field

$$\boldsymbol{u}_t^Y(\boldsymbol{y} \mid \boldsymbol{z}^X) \triangleq \boldsymbol{A} \boldsymbol{u}_t^X(\boldsymbol{x} \mid \boldsymbol{z}^X) - (a_t a_t' \sigma_0^2 + b_t b_t' \sigma_1^2) \nabla_{\boldsymbol{y}} \log p_t^Y(\boldsymbol{y} \mid \boldsymbol{z}^X) \tag{16}$$

generates the conditional probability path $p_t^Y(\boldsymbol{y} \mid \boldsymbol{z}^X)$. Taking the expectation of Eq. (16) over $q_t(\boldsymbol{z}^X \mid \boldsymbol{y})$ and leveraging Eq. (9), we can obtain the marginal vector field in the $\mathcal{Y}$-space in terms of the optimal PCFM solution and the score function, as described in the following lemma.

**Lemma 1.** *The $\mathcal{Y}$-space marginal vector field that generates the probability path $p_t^Y$ takes the form*

$$\boldsymbol{u}_t^Y(\boldsymbol{y}) = \boldsymbol{A} \boldsymbol{v}_{\boldsymbol{\theta}^*}^X(\boldsymbol{y}, t) - (a_t a_t' \sigma_0^2 + b_t b_t' \sigma_1^2) \nabla_{\boldsymbol{y}} \log p_t^Y(\boldsymbol{y}). \tag{17}$$

Meanwhile, the relationship between the $\mathcal{Y}$-space marginal vector field $\boldsymbol{u}_t^Y(\boldsymbol{y})$ and the score function $\nabla_{\boldsymbol{y}} \log p_t^Y(\boldsymbol{y})$ is established by the following lemma.

**Lemma 2.** *Note that $p_1^Y(\boldsymbol{y}) = \int p_1(\boldsymbol{y} \mid \boldsymbol{x}) p_1^X(\boldsymbol{x}) \mathrm{d}\boldsymbol{x} = \mathcal{CN}(\boldsymbol{y} \mid \boldsymbol{0}, 2\boldsymbol{A}\boldsymbol{A}^* + \sigma_1^2 \boldsymbol{I}_{Cd})$. Then,*

$$\boldsymbol{u}_t^Y(\boldsymbol{y}) = \frac{a_t'}{a_t} \boldsymbol{y} - b_t \left( b_t' - \frac{a_t'}{a_t} b_t \right) (2\boldsymbol{A}\boldsymbol{A}^* + \sigma_1^2 \boldsymbol{I}_{Cd}) \nabla_{\boldsymbol{y}} \log p_t^Y(\boldsymbol{y}). \tag{18}$$

The proof is done by writing $\boldsymbol{u}_t^Y(\boldsymbol{y})$ and $\nabla_{\boldsymbol{y}} \log p_t^Y(\boldsymbol{y})$ in terms of the conditional expectation $\mathbb{E}_{q_t(\boldsymbol{y}_0 \mid \boldsymbol{y})}[\boldsymbol{y}_1]$, which can be found in Appendix A. Combining Lemma 1 and Lemma 2 by canceling out the score function gives the following proposition that relates the $\mathcal{Y}$-space marginal vector field to the optimal solution to PCFM.

**Proposition 3 (Vector fields under projection).** *For $a_t = 1 - t$ and $b_t = t$, the $\mathcal{Y}$-space marginal vector field $\boldsymbol{u}_t^Y(\boldsymbol{y})$ can be expressed by $\boldsymbol{v}_{\boldsymbol{\theta}^*}^X(\boldsymbol{y}, t)$ as*

$$\boldsymbol{u}_t^Y(\boldsymbol{y}) = \boldsymbol{A} \boldsymbol{v}_{\boldsymbol{\theta}^*}^X(\boldsymbol{y}, t) - \frac{c_t}{1-t} \left[ (c_t + \sigma_1^2) \boldsymbol{I}_{Cd} + 2\boldsymbol{A}\boldsymbol{A}^* \right]^{-1} \left[ (1-t) \boldsymbol{A} \boldsymbol{v}_{\boldsymbol{\theta}^*}^X(\boldsymbol{y}, t) + \boldsymbol{y} \right], \tag{19}$$

*where $c_t \triangleq (1 - t) \left( \frac{1-t}{t} \sigma_0^2 - \sigma_1^2 \right)$. In addition, left-multiplying both sides with $\boldsymbol{A}^*$ gives the more computationally friendly formula when $Cd > D$:*

$$\boldsymbol{A}^* \boldsymbol{u}_t^Y(\boldsymbol{y}) = \boldsymbol{A}^* \boldsymbol{A} \boldsymbol{v}_{\boldsymbol{\theta}^*}^X(\boldsymbol{y}, t) - \frac{c_t}{1-t} \left[ (c_t + \sigma_1^2) \boldsymbol{I}_D + 2\boldsymbol{A}^* \boldsymbol{A} \right]^{-1} \boldsymbol{A}^* \left[ (1-t) \boldsymbol{A} \boldsymbol{v}_{\boldsymbol{\theta}^*}^X(\boldsymbol{y}, t) + \boldsymbol{y} \right]. \tag{20}$$

---

**Algorithm 1:** Reconstruction via Dual-Space Cyclic Integration with PCFM

---

**Input:** k-space measurement $\boldsymbol{y}_0$, pretrained optimal solution to PCFM $\boldsymbol{v}_{\boldsymbol{\theta}^*}^X(\cdot, t)$, number of time steps $T$.

**Output:** Reconstructed image $\boldsymbol{x}_0$ of $\boldsymbol{y}_0$.

1 **for** $t = 0, \ldots, {}^{(T-1)}/T$ **do**

2     $\boldsymbol{y}_{t+1/T} \leftarrow \boldsymbol{y}_t + \frac{1}{T}\boldsymbol{u}_t^Y(\boldsymbol{y})$               ▷ Forward integration using Proposition 3

3 Sample $\boldsymbol{x}_1 \sim p_1(\boldsymbol{x} \mid \boldsymbol{y})$.             ▷ Posterior sampling with Eq. (22)

4 **for** $t \in \{{}^{(T-1)}/T, \ldots, 0\}$ **do**

5     $\widetilde{\boldsymbol{y}}_{t+1/T} \leftarrow a_{t+1/T}\boldsymbol{y}_0 + b_{t+1/T}\boldsymbol{y}_1$

6     $\widetilde{\boldsymbol{x}}_t \leftarrow \boldsymbol{x}_{t+1/T} - \frac{1}{T}\boldsymbol{v}_{\boldsymbol{\theta}^*}^X(\widetilde{\boldsymbol{y}}_{t+1/T}, t + {}^1/T)$       ▷ Backward integration

7     $\boldsymbol{x}_t \leftarrow \widetilde{\boldsymbol{x}}_t - \boldsymbol{A}^+(\boldsymbol{A}\widetilde{\boldsymbol{x}}_t - \widetilde{\boldsymbol{y}}_t)$           ▷ Data consistency step

8 **return** $\boldsymbol{x}_0$.

---

This proposition asserts that, using the optimal solution of the PCFM, an associated marginal vector field can be derived, which facilitates the generation of the probability flow within the measurement space. Consequently, to sample from $p(\boldsymbol{y}_1 \mid \boldsymbol{y}_0)$, the $\mathcal{Y}$-space flow can be forward integrated using $\boldsymbol{u}_t^Y$ to produce the subsequent $\boldsymbol{y}_1$ from the observed initial $\boldsymbol{y}_0$, i.e.,

$$\mathrm{d}\boldsymbol{y}_t = \boldsymbol{u}_t^Y(\boldsymbol{y}_t)\mathrm{d}t. \tag{21}$$

**Posterior sampling.** The posterior $p(\boldsymbol{x}_1 \mid \boldsymbol{y}_1)$ follows a closed-form Gaussian distribution

$$
\begin{aligned}
p(\boldsymbol{x}_1 \mid \boldsymbol{y}_1) = p_1(\boldsymbol{x} \mid \boldsymbol{y}) &\propto p_1^X(\boldsymbol{x})p_1(\boldsymbol{y} \mid \boldsymbol{x}) \\
&= \mathcal{CN}\left(\boldsymbol{x} \,\middle|\, \left(\frac{\sigma_1^2}{2}\boldsymbol{I}_D + \boldsymbol{A}^*\boldsymbol{A}\right)^{-1}\boldsymbol{A}^*\boldsymbol{y}, \sigma_1^2\left(\frac{\sigma_1^2}{2}\boldsymbol{I}_D + \boldsymbol{A}^*\boldsymbol{A}\right)^{-1}\right),
\end{aligned}
\tag{22}
$$

which can be easily sampled by linear transformation of a standard Gaussian vector.

**Backward integration and measurement consistency.** The distribution $p(\boldsymbol{x}_0 \mid \boldsymbol{x}_1)$ is a delta distribution given $\boldsymbol{x}_1$ due to the $\mathcal{X}$-space flow, which we can approximate using the backward ODE integration with the PCFM optimal solution, i.e.,

$$\mathrm{d}\widetilde{\boldsymbol{x}}_t = \boldsymbol{v}_{\boldsymbol{\theta}^*}^X(\widetilde{\boldsymbol{y}}_t, t)\mathrm{d}t, \tag{23}$$

where $\widetilde{\boldsymbol{x}}_1 \triangleq \boldsymbol{x}_1$ and $\widetilde{\boldsymbol{y}}_t \triangleq a_t\boldsymbol{y}_0 + b_t\boldsymbol{y}_1$. In addition, to enforce measurement consistency with $\widetilde{\boldsymbol{y}}_t$ as in many inverse problem solving algorithms (Daras et al., 2024a), we can employ the range-null decomposition (Wang et al., 2023) after each step of the backward integration if the observed measurement $\boldsymbol{y}_0$ is clean, i.e.,

$$\boldsymbol{x}_t \leftarrow \widetilde{\boldsymbol{x}}_t - \boldsymbol{A}^+(\boldsymbol{A}\widetilde{\boldsymbol{x}}_t - \widetilde{\boldsymbol{y}}_t), \tag{24}$$

where the pseudoinverse operator $\boldsymbol{A}^+$ can be approximated by the CG method. However, if the measurement $\boldsymbol{y}_0$ is noisy, this step will also inject noise into the reconstruction. To address this, we find that the Plug-and-Play (PnP) framework can perform better on noisy measurement (Combettes & Wajs, 2005; Venkatakrishnan et al., 2013; Martin et al., 2025), which we introduce in Appendix B. Ablation study on the backward sampling strategies is presented in Appendix E.2.

**Reconstruction algorithm.** The overall discrete-time algorithm is outlined in Alg. 1 for scenarios involving noiseless measurements and in Alg. 2 within Appendix B for noisy measurements.

## 4 RELATED WORK

**Diffusion model & flow matching for inverse problems.** A comprehensive review on diffusion models for inverse problems is provided by (Daras et al., 2024a) and (Chung et al., 2025). For example, diffusion posterior sampling (DPS) is proposed to sample from the posterior distribution based on score matching (Chung et al., 2023) which however requires the number of function evaluations (NFEs)=1000. (Wang et al., 2023) proposed range-null decomposition based on DDIM sampling

(Song et al., 2021), which only needs NFEs=100. Recently, flow matching has also been explored to solve inverse problems (Pokle et al., 2024; Zhang et al., 2024; Qin et al., 2025; Martin et al., 2025; Yan et al., 2025). For example, (Pokle et al., 2024) proposed to combine the prior score function based on flow matching and the likelihood score based on ΠGDM (Song et al., 2023). (Martin et al., 2025) proposed to integrate the PnP framework with flow matching. Nevertheless, existing diffusion models and flow matching require large amounts of ground-truth data to learn their prior distribution, which are impossible or expensive to acquire for MRI in real scenarios, for example, in dynamic or low-field MRI (Lustig et al., 2007; Marques et al., 2019).

**Self-supervised & unsupervised MRI reconstruction.** We explicitly distinguish between self-supervised methods based on additional subsampling of the available k-space and unsupervised approaches that use all the observed undersampled k-space measurement, while both are referred to as self-supervised in a recent benchmark study (Wang et al., 2025). Multiple recent studies follow the prior learning paradigm and have explored methodologies to infer the prior distribution of ground-truth data from only corrupted measurements based on diffusion models or flow matching. The ambient diffusion family (Daras et al., 2023; Aali et al., 2025; Daras et al., 2024b; 2025) proposed optimizing the denoising diffusion objective through a self-supervised approach by introducing further corruption to the measurements. Our work is more related to (Kawar et al., 2024) and (Luo et al., 2025) where they proposed adapting the Ensemble SURE frameworks (Aggarwal et al., 2022) to the diffusion model and flow matching objectives in an unsupervised fashion. They focused solely on a single-coil MRI model with a feasible weighted projection operator. In contrast, we introduce a rigorous GSURE-based projected CFM formulation that facilitates optimization in parallel MRI.

## 5 NUMERICAL EXPERIMENTS

### 5.1 EXPERIMENTAL SETUPS

**Datasets and preprocessing.** Experiments were conducted on the NYU fastMRI (Knoll et al., 2020; Zbontar et al., 2018) and the CMRxRecon challenge (2023) (Wang et al., 2024; Lyu et al., 2025) datasets to evaluate the performance of the model for accelerated multi-coil MRI reconstruction. A selection of 11094/1584/3172 T2-weighted brain MRI slices and 5451/779/1557 cardiac T1/T2 quantitative mapping slices was randomly sampled for training/validation/test, respectively. Ground-truth images were obtained by the SENSE reconstruction from fully sampled k-space (Pruessmann et al., 1999), i.e., $x_0 \triangleq (\sum_c S_c^* S_c)^{-1} \sum_c S_c^* F^* \widehat{y}_{0,c}$, where $\widehat{y}_0$ is the fully sampled k-space measurement, and the coil sensitivity maps were estimated by ESPIRiT (Uecker et al., 2014). We retrospectively simulated a random Cartesian (1D) undersampling mask for each image, where every mask contains fully sampled low-frequency k-space lines. The other lines were randomly uniformly sampled according to the acceleration factor. The zero-filled adjoint transform $A^* y_0$ is the undersampled image before reconstruction.

**Implementation details.** The conditional OT path (Lipman et al., 2023) is adopted for the proposed framework, where $a_t = t$ and $b_t = 1 - t$. The noise level of the original k-space is assumed to be $\sigma_0 = 10^{-3}$, which can be considered as noiseless. We set $\sigma_1 = 0$ for simplicity. We use ADM U-Net (Dhariwal & Nichol, 2021) as the network architecture for velocity field prediction, where each intermediate convolutional block is followed by adaptive group normalization whose parameters are conditioned on the time points. Multi-head attention and dropout layers are applied at the lowest three resolutions of the network. We train the network from scratch on the training data by the AdamW optimizer (Loshchilov & Hutter, 2019) for 100K steps, with a learning rate of $10^{-4}$ and a weight decay coefficient of $0.1$. Exponential moving average of the network parameters was performed every 100 training steps with a rate of 0.99. In the inference phase, we set the number of time steps as $T = 10$ and the number of CG steps (Appendix C.1) for solving $A^+$ as 30.

### 5.2 BENCHMARK STUDY ON PUBLIC DATASETS

We benchmark our method against three types of baseline approaches on the fastMRI brain and CMRxRecon datasets. (A) Supervised methods that require fully sampled MRIs during training: **MoDL** (Aggarwal et al., 2018), **DDNM⁺** (Wang et al., 2023), **OT-ODE** (Pokle et al., 2024), and **PnP-Flow**, (Martin et al., 2025). (B) Self-supervised methods that learn to reconstruct by additional subsampling of the available k-space measurements: **SSDU** (Yaman et al., 2020), **Weighted SSDU** (Millard & Chiew, 2023) and **Robust SSDU** (Millard & Chiew, 2024). (C) Unsupervised methods

Table 1: Qualitative results of $4\times$ and $8\times$ accelerated multi-coil MRI reconstruction using various reconstruction methods on the fastMRI brain data with the random uniform undersampling pattern. Best results within each supervision category are highlighted in **bold**. The difference in metrics is statistically significant between our method and the others by the two-sided paired $t$-test ($p < 0.05$).

| Training Supervision | Reconstruction Method | SSIM ↑ | | PSNR ↑ | | NFEs ↓ |
|---|---|---|---|---|---|---|
| | | $4\times$ | $8\times$ | $4\times$ | $8\times$ | |
| None | Zero-Filled | $0.815 \pm 0.087$ | $0.730 \pm 0.113$ | $28.28 \pm 3.80$ | $24.44 \pm 3.98$ | N/A |
| Supervised | MoDL | $\mathbf{0.970 \pm 0.036}$ | $0.916 \pm 0.044$ | $39.71 \pm 2.93$ | $32.32 \pm 3.07$ | 1 |
| | DDNM$^+$ | $0.938 \pm 0.041$ | $\mathbf{0.920 \pm 0.040}$ | $\mathbf{40.00 \pm 2.78}$ | $\mathbf{35.24 \pm 2.85}$ | 100 |
| | OT-ODE | $0.907 \pm 0.054$ | $0.852 \pm 0.060$ | $33.95 \pm 2.32$ | $28.86 \pm 2.94$ | 100 |
| | PnP-Flow | $0.951 \pm 0.044$ | $0.913 \pm 0.043$ | $37.92 \pm 2.44$ | $32.85 \pm 2.80$ | 100 |
| Self-supervised | SSDU | $0.831 \pm 0.076$ | $0.792 \pm 0.094$ | $28.13 \pm 2.44$ | $26.61 \pm 3.68$ | 1 |
| | Weighted SSDU | $0.939 \pm 0.049$ | $\mathbf{0.870 \pm 0.055}$ | $36.09 \pm 2.64$ | $\mathbf{29.72 \pm 2.90}$ | 1 |
| | Robust SSDU | $\mathbf{0.941 \pm 0.047}$ | $0.856 \pm 0.063$ | $\mathbf{36.23 \pm 2.58}$ | $29.72 \pm 2.90$ | 1 |
| Unsupervised | REI | $0.683 \pm 0.120$ | $0.715 \pm 0.118$ | $21.59 \pm 2.90$ | $21.62 \pm 2.76$ | 1 |
| | MOI | $0.869 \pm 0.065$ | $0.747 \pm 0.105$ | $30.97 \pm 2.74$ | $25.16 \pm 3.94$ | 1 |
| | ENSURE | $0.899 \pm 0.065$ | $0.800 \pm 0.104$ | $32.75 \pm 4.20$ | $27.12 \pm 4.33$ | 1 |
| | GTF$^2$M | $0.916 \pm 0.055$ | $0.852 \pm 0.057$ | $33.99 \pm 2.33$ | $28.40 \pm 3.05$ | 20 |
| | PCFM (Ours) | $\mathbf{0.983 \pm 0.032}$ | $\mathbf{0.948 \pm 0.034}$ | $\mathbf{42.19 \pm 3.71}$ | $\mathbf{35.08 \pm 3.35}$ | 20 |

Table 2: Qualitative results of $4\times$ and $8\times$ accelerated multi-coil MRI reconstruction using various reconstruction methods on the CMRxRecon 2023 cardiac T1/T2 quantitative mapping data with the random uniform undersampling pattern. Best results within each supervision category are highlighted in **bold**. The difference in metrics is statistically significant between our method and the others by the two-sided paired $t$-test ($p < 0.05$).

| Training Supervision | Reconstruction Method | SSIM ↑ | | PSNR ↑ | | NFEs ↓ |
|---|---|---|---|---|---|---|
| | | $4\times$ | $8\times$ | $4\times$ | $8\times$ | |
| None | Zero-Filled | $0.769 \pm 0.057$ | $0.747 \pm 0.056$ | $27.01 \pm 1.97$ | $26.11 \pm 1.90$ | N/A |
| Supervised | MoDL | $\mathbf{0.979 \pm 0.011}$ | $0.943 \pm 0.025$ | $41.79 \pm 3.38$ | $36.36 \pm 3.14$ | 1 |
| | DDNM$^+$ | $0.977 \pm 0.011$ | $\mathbf{0.953 \pm 0.022}$ | $\mathbf{44.93 \pm 3.35}$ | $\mathbf{39.60 \pm 3.31}$ | 100 |
| | OT-ODE | $0.922 \pm 0.036$ | $0.870 \pm 0.052$ | $35.65 \pm 3.23$ | $32.08 \pm 3.00$ | 100 |
| | PnP-Flow | $0.963 \pm 0.021$ | $0.924 \pm 0.038$ | $39.77 \pm 3.57$ | $35.13 \pm 3.46$ | 100 |
| Self-supervised | SSDU | $0.888 \pm 0.035$ | $0.811 \pm 0.051$ | $32.64 \pm 2.59$ | $29.23 \pm 2.37$ | 1 |
| | Weighted SSDU | $0.872 \pm 0.050$ | $0.831 \pm 0.050$ | $32.11 \pm 2.88$ | $29.82 \pm 2.58$ | 1 |
| | Robust SSDU | $\mathbf{0.912 \pm 0.031}$ | $\mathbf{0.861 \pm 0.044}$ | $\mathbf{33.93 \pm 2.59}$ | $\mathbf{31.10 \pm 2.66}$ | 1 |
| Unsupervised | REI | $0.736 \pm 0.056$ | $0.716 \pm 0.060$ | $23.65 \pm 2.12$ | $26.62 \pm 1.96$ | 1 |
| | MOI | $0.971 \pm 0.015$ | $0.874 \pm 0.040$ | $40.13 \pm 3.26$ | $31.47 \pm 2.62$ | 1 |
| | ENSURE | $0.918 \pm 0.028$ | $0.849 \pm 0.052$ | $34.57 \pm 3.13$ | $30.27 \pm 2.48$ | 1 |
| | GTF$^2$M | $0.918 \pm 0.028$ | $0.881 \pm 0.038$ | $33.67 \pm 2.43$ | $31.33 \pm 2.42$ | 20 |
| | PCFM (Ours) | $\mathbf{0.994 \pm 0.008}$ | $\mathbf{0.974 \pm 0.016}$ | $\mathbf{52.39 \pm 9.81}$ | $\mathbf{40.50 \pm 3.84}$ | 20 |

that learn to reconstruct using all available k-space measurements: **REI** (Chen et al., 2022), **MOI** (Tachella et al., 2022), **ENSURE** (Aggarwal et al., 2022), and **GTF$^2$M** (Luo et al., 2025). Details of the baseline setups can be found in Appendix D.

Table 1 and Table 2 present quantitative results on the test data of multi-coil brain and cardiac MRI, respectively. The tables show that our method ranks first in terms of SSIM and PSNR on both datasets among all self-supervised and unsupervised methods, and outperforms all supervised approaches except for PSNR on fastMRI $8\times$ data. Furthermore, our approach demonstrates better efficiency relative to baseline approaches based on supervised diffusion models or flow matching, even though it employs the same network structure and training strategy. This is achieved with a significant decrease in NFEs when compared to supervised diffusion models and flow matching-based techniques. Fig. 3 and Fig. S1 visualize reconstruction and error map on several multi-coil brain and cardiac MRI test samples, respectively. Our method produces error maps that either match or exceed those generated by supervised baseline approaches, particularly in areas around anatomical boundaries where high-frequency details are absent in the undersampled k-space. Additional ablation studies on forward/backward sampling strategies and inference time comparison are provided in Appendix E.

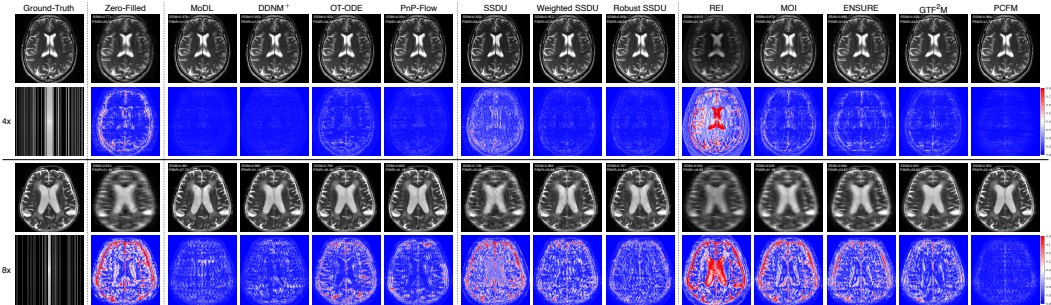

Figure 3: Visualization of reconstruction on two test samples of $4\times$ and $8\times$ accelerated multi-coil brain MRI from the compared methods. The k-space are presented in log-scale absolute values. The error maps are presented in values relative to the peak intensity in the ground-truth image.

Table 3: Qualitative results of $4\times$ and $8\times$ reconstruction of CMRxRecon 2023 cardiac T1/T2 quantitative mapping images trained and tested both on noisy data. The difference in metrics is statistically significant between our method and the others by the two-sided paired $t$-test ($p < 0.05$).

| Training Noise level | Reconstruction Method | SSIM ↑ | | PSNR ↑ | | NFEs ↓ |
|---|---|---|---|---|---|---|
| | | $4\times$ | $8\times$ | $4\times$ | $8\times$ | |
| $\sigma_0 = 0.05$ | Zero-Filled | $0.768 \pm 0.057$ | $0.746 \pm 0.056$ | $27.01 \pm 1.97$ | $26.11 \pm 1.90$ | N/A |
| | Robust SSDU | $0.912 \pm 0.031$ | $0.855 \pm 0.045$ | $34.06 \pm 2.69$ | $30.81 \pm 2.61$ | 10 |
| | ENSURE | $0.861 \pm 0.050$ | $0.800 \pm 0.058$ | $33.25 \pm 2.75$ | $29.80 \pm 2.39$ | 1 |
| | PnP-PCFM (Ours) | $\mathbf{0.928 \pm 0.027}$ | $\mathbf{0.892 \pm 0.038}$ | $\mathbf{35.58 \pm 2.75}$ | $\mathbf{33.14 \pm 2.82}$ | 20 |
| $\sigma_0 = 0.1$ | Zero-Filled | $0.763 \pm 0.058$ | $0.744 \pm 0.057$ | $26.98 \pm 1.97$ | $26.10 \pm 1.90$ | N/A |
| | Robust SSDU | $0.906 \pm 0.033$ | $0.855 \pm 0.045$ | $33.80 \pm 2.68$ | $30.91 \pm 2.64$ | 10 |
| | ENSURE | $0.784 \pm 0.063$ | $0.769 \pm 0.062$ | $31.39 \pm 2.54$ | $28.85 \pm 2.30$ | 1 |
| | PnP-PCFM (Ours) | $\mathbf{0.921 \pm 0.029}$ | $\mathbf{0.889 \pm 0.039}$ | $\mathbf{35.62 \pm 2.85}$ | $\mathbf{33.04 \pm 2.82}$ | 20 |
| $\sigma_0 = 0.2$ | Zero-Filled | $0.744 \pm 0.062$ | $0.732 \pm 0.060$ | $26.89 \pm 1.96$ | $26.05 \pm 1.91$ | N/A |
| | Robust SSDU | $0.878 \pm 0.039$ | $0.822 \pm 0.049$ | $32.47 \pm 2.52$ | $29.78 \pm 2.39$ | 10 |
| | ENSURE | $0.563 \pm 0.094$ | $0.542 \pm 0.090$ | $25.88 \pm 2.55$ | $25.29 \pm 2.39$ | 1 |
| | PnP-PCFM (Ours) | $\mathbf{0.885 \pm 0.038}$ | $\mathbf{0.868 \pm 0.044}$ | $\mathbf{34.34 \pm 2.76}$ | $\mathbf{32.52 \pm 2.77}$ | 20 |

## 5.3 NOISY DATA

We proceed to assess our method on data with noise by introducing additive Gaussian noise with $\sigma_0 = 0.05, 0.1, 0.2$ to the training set based on the original k-space measurements. The model's performance is evaluated on both noiseless and noisy test datasets. For evaluation, Alg. 1 is used if the data set is noiseless, otherwise Alg. 2 is used. Table S4 displays the quantitative results regarding the noiseless test data derived from the CMRxRecon 2023 dataset. A notable observation is that the model retains its performance as if the training data were free of noise, suggesting that PCFM effectively learns the prior distribution of fully sampled MRI even when trained on noisy and undersampled inputs. Table 3 presents the quantitative results for noisy test data, comparing PnP-PCFM against the baseline models Robust SSDU and ENSURE, which are designed to handle noisy inputs. The proposed approach exhibits superior performance compared to them. Fig. S2 provides a visualization of reconstruction and error maps for two test samples involving $4\times$ and $8\times$ accelerated multi-coil cardiac MRI with varying levels of additive Gaussian noise.

## 6 CONCLUSION

In this study, we present Projected Conditional Flow Matching (PCFM), a new framework that utilizes the generalized Stein's unbiased risk estimator to learn the prior distribution of fully sampled parallel MRI using solely undersampled k-space data. From the optimal PCFM solution, we derived a marginal vector field within the associated measurement space, facilitating the creation of a dual-space cyclic integration method for MRI reconstruction. Experimental assessments on two parallel MRI datasets demonstrate that PCFM attains leading performance relative to prior baselines on both noiseless and noisy data.

DECLARATIONS

**LLM Usage:** We note that large language models (LLMs) were used solely to improve language clarity; all scientific content and ideas were developed by the authors.

**Ethics Statement:** This research complies with the ICLR Code of Ethics. The study uses multiple datasets: publicly available fastMRI and CMRxRecon 2023. All datasets are de-identified and contain no personally identifiable information. No direct human subject recruitment was involved, and the work does not raise foreseeable risks to participants.

**Reproducibility Statement:** We provide detailed descriptions of datasets, preprocessing steps, model architectures, training procedures, and evaluation protocols in Section 3 and Section 5 of the manuscript and Appendix B, Appendix C, and Appendix D of the supplementary material to ensure reproducibility of our model and compared approaches. Source code and model weights will be made publicly available at https://github.com/anonymous upon acceptance.

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

# Appendices

## A    PROOFS

**Proposition 1** (**Optimal solution to PCFM**). *The minimizer of the PCFM objective is given by*

$$\boldsymbol{v}_{\boldsymbol{\theta}^*}^X(\boldsymbol{y}, t) = \mathbb{E}_{q_t(\boldsymbol{z}^X|\boldsymbol{y}), p_t^X(\boldsymbol{x}|\boldsymbol{z}^X)} \left[ \boldsymbol{u}_t^X(\boldsymbol{x} \mid \boldsymbol{z}^X) \right] + \boldsymbol{w}, \tag{S1}$$

*where* $q_t(\boldsymbol{z}^X \mid \boldsymbol{y}) = \frac{q(\boldsymbol{z}^X) p_t^Y(\boldsymbol{y}|\boldsymbol{z}^X)}{p_t^Y(\boldsymbol{y})}$, *and* $\boldsymbol{w}$ *is any vector in the null space of* $\boldsymbol{A}$, *i.e.,* $\boldsymbol{A}\boldsymbol{w} = \boldsymbol{0}$. *In particular, when* $\boldsymbol{u}_t^X(\boldsymbol{x} \mid \boldsymbol{z}^X) = a_t' \boldsymbol{x}_0 + b_t' \boldsymbol{x}_1$ *that is independent of* $\boldsymbol{x}$, *we have*

$$\boldsymbol{v}_{\boldsymbol{\theta}^*}^X(\boldsymbol{y}, t) = \mathbb{E}_{q_t(\boldsymbol{z}^X|\boldsymbol{y})} \left[ \boldsymbol{u}_t^X(\boldsymbol{x} \mid \boldsymbol{z}^X) \right] + \boldsymbol{w}. \tag{S2}$$

*Proof.* Since $q(\boldsymbol{z}^X) p_t^X(\boldsymbol{x} \mid \boldsymbol{z}^X) p_t^Y(\boldsymbol{y} \mid \boldsymbol{z}^X) = p_t^Y(\boldsymbol{y}) q_t^Y(\boldsymbol{z}^X \mid \boldsymbol{y}) p_t^X(\boldsymbol{x} \mid \boldsymbol{z}^X)$ and by the law of total expectation, the PCFM objective can be written as

$$\begin{aligned}
\mathcal{L}_{\mathrm{PCFM}}(\boldsymbol{\theta}) &\triangleq \mathbb{E}_{t, q(\boldsymbol{z}^X), p_t^X(\boldsymbol{x}|\boldsymbol{z}^X), p_t^Y(\boldsymbol{y}|\boldsymbol{z}^X)} \left\| \boldsymbol{P} \left[ \boldsymbol{v}_{\boldsymbol{\theta}}^X(\boldsymbol{y}, t) - \boldsymbol{u}_t^X(\boldsymbol{x} \mid \boldsymbol{z}^X) \right] \right\|_2^2 \\
&= \mathbb{E}_{t, p_t^Y(\boldsymbol{y})} \mathbb{E}_{q_t^Y(\boldsymbol{z}^X|\boldsymbol{y}), p_t^X(\boldsymbol{x}|\boldsymbol{z}^X)} \left\| \boldsymbol{P} \left[ \boldsymbol{v}_{\boldsymbol{\theta}}^X(\boldsymbol{y}, t) - \boldsymbol{u}_t^X(\boldsymbol{x} \mid \boldsymbol{z}^X) \right] \right\|_2^2.
\end{aligned} \tag{S3}$$

To minimize the total expectation, we can minimize the inner conditional expectation for each value of $t$ and $\boldsymbol{y}$ independently. Let $\widehat{\boldsymbol{u}}_t^X(\boldsymbol{y}) \triangleq \mathbb{E}_{q_t(\boldsymbol{z}^X|\boldsymbol{y}), p_t^X(\boldsymbol{x}|\boldsymbol{z}^X)} \left[ \boldsymbol{u}_t^X(\boldsymbol{x} \mid \boldsymbol{z}^X) \right]$. For fixed $t$ and $\boldsymbol{y}$, the inner expectation can be transformed as

$$\begin{aligned}
\boldsymbol{I}_{t,\boldsymbol{y}}(\boldsymbol{\theta}) &\triangleq \mathbb{E}_{q_t^Y(\boldsymbol{z}^X|\boldsymbol{y}), p_t^X(\boldsymbol{x}|\boldsymbol{z}^X)} \left\| \boldsymbol{P} \left[ \boldsymbol{v}_{\boldsymbol{\theta}}^X(\boldsymbol{y}, t) - \boldsymbol{u}_t^X(\boldsymbol{x} \mid \boldsymbol{z}^X) \right] \right\|_2^2 \\
&= \mathbb{E}_{q_t^Y(\boldsymbol{z}^X|\boldsymbol{y}), p_t^X(\boldsymbol{x}|\boldsymbol{z}^X)} \left\| \boldsymbol{P} \left[ \boldsymbol{v}_{\boldsymbol{\theta}}^X(\boldsymbol{y}, t) - \widehat{\boldsymbol{u}}_t^X(\boldsymbol{y}) + \widehat{\boldsymbol{u}}_t^X(\boldsymbol{y}) - \boldsymbol{u}_t^X(\boldsymbol{x} \mid \boldsymbol{z}^X) \right] \right\|_2^2 \\
&= \mathbb{E}_{q_t^Y(\boldsymbol{z}^X|\boldsymbol{y}), p_t^X(\boldsymbol{x}|\boldsymbol{z}^X)} \Big[ \left\| \boldsymbol{P} \left[ \boldsymbol{v}_{\boldsymbol{\theta}}^X(\boldsymbol{y}, t) - \widehat{\boldsymbol{u}}_t^X(\boldsymbol{y}) \right] \right\|_2^2 \\
&\quad + 2 \left( \boldsymbol{v}_{\boldsymbol{\theta}}^X(\boldsymbol{y}, t) - \widehat{\boldsymbol{u}}_t^X(\boldsymbol{y}) \right)^* \boldsymbol{P} \left( \widehat{\boldsymbol{u}}_t^X(\boldsymbol{y}) - \boldsymbol{u}_t^X(\boldsymbol{x} \mid \boldsymbol{z}^X) \right) \Big] + \mathrm{const.},
\end{aligned} \tag{S4}$$

where we note that

$$\begin{aligned}
&\mathbb{E}_{q_t^Y(\boldsymbol{z}^X|\boldsymbol{y}), p_t^X(\boldsymbol{x}|\boldsymbol{z}^X)} \left( \boldsymbol{v}_{\boldsymbol{\theta}}^X(\boldsymbol{y}, t) - \widehat{\boldsymbol{u}}_t^X(\boldsymbol{y}) \right)^* \boldsymbol{P} \left( \widehat{\boldsymbol{u}}_t^X(\boldsymbol{y}) - \boldsymbol{u}_t^X(\boldsymbol{x} \mid \boldsymbol{z}^X) \right) \\
&= \left( \boldsymbol{v}_{\boldsymbol{\theta}}^X(\boldsymbol{y}, t) - \widehat{\boldsymbol{u}}_t^X(\boldsymbol{y}) \right)^* \boldsymbol{P} \left[ \widehat{\boldsymbol{u}}_t^X(\boldsymbol{y}) - \mathbb{E}_{q_t^Y(\boldsymbol{z}^X|\boldsymbol{y}), p_t^X(\boldsymbol{x}|\boldsymbol{z}^X)} \boldsymbol{u}_t^X(\boldsymbol{x} \mid \boldsymbol{z}^X) \right] = 0.
\end{aligned} \tag{S5}$$

Therefore, $\boldsymbol{I}_{t,\boldsymbol{y}}(\boldsymbol{\theta}) = \mathbb{E}_{q_t^Y(\boldsymbol{z}^X|\boldsymbol{y}), p_t^X(\boldsymbol{x}|\boldsymbol{z}^X)} \left\| \boldsymbol{P} \left[ \boldsymbol{v}_{\boldsymbol{\theta}}^X(\boldsymbol{y}, t) - \widehat{\boldsymbol{u}}_t^X(\boldsymbol{y}) \right] \right\|_2^2$, which is minimized when

$$\boldsymbol{v}_{\boldsymbol{\theta}}^X(\boldsymbol{y}, t) = \widehat{\boldsymbol{u}}_t^X(\boldsymbol{y}) + \boldsymbol{w}, \tag{S6}$$

where $\boldsymbol{w}$ is in the null space of $\boldsymbol{P}$, i.e., $\boldsymbol{P}\boldsymbol{w} = \boldsymbol{0}$, which is equivalent to $\boldsymbol{A}\boldsymbol{w} = \boldsymbol{0}$.    $\square$

**Proposition 2** (**Unsupervised transformation of PCFM**). *Assuming deterministic conditional probability paths* $\boldsymbol{x} = a_t \boldsymbol{x}_0 + b_t \boldsymbol{x}_1$ *and* $\boldsymbol{y} = a_t \boldsymbol{y}_0 + b_t \boldsymbol{y}_1$ *with* $\boldsymbol{y}_0 = \boldsymbol{A}\boldsymbol{x}_0 + \boldsymbol{e}_0$ *and* $\boldsymbol{y}_1 = \boldsymbol{A}\boldsymbol{x}_1 + \boldsymbol{e}_1$, *where* $\boldsymbol{e}_0 \sim \mathcal{CN}(\boldsymbol{0}, \sigma_0^2 \boldsymbol{I}_{Cd})$ *and* $\boldsymbol{e}_1 \sim \mathcal{CN}(\boldsymbol{0}, \sigma_1^2 \boldsymbol{I}_{Cd})$, *then up to a constant the PCFM objective can be transformed to*

$$\mathbb{E}_{t, q(\boldsymbol{z}^Y), p_t^Y(\boldsymbol{y}|\boldsymbol{z}^Y)} \left[ \left\| \boldsymbol{P} \left[ \boldsymbol{v}_{\boldsymbol{\theta}}^X(\boldsymbol{A}^* \boldsymbol{y}, t) - \widehat{\boldsymbol{u}}_{t,ML}^X \right] \right\|_2^2 + \frac{2a_t}{a_t'} [(a_t' \sigma_0)^2 + (b_t' \sigma_1)^2] \nabla_{\boldsymbol{A}^* \boldsymbol{y}} \cdot \boldsymbol{P} \boldsymbol{v}_{\boldsymbol{\theta}}^X(\boldsymbol{A}^* \boldsymbol{y}, t) \right], \tag{S7}$$

*where* $q(\boldsymbol{z}^Y) = q(\boldsymbol{y}_0) q(\boldsymbol{y}_1) = q(\boldsymbol{y}_0) \mathbb{E}_{q(\boldsymbol{x}_1)} \left[ p(\boldsymbol{y}_1 \mid \boldsymbol{x}_1) \right]$ *is sampled by the MRI and Monte Carlo estimation,* $\boldsymbol{P} = \boldsymbol{A}^+ \boldsymbol{A}$ *is the range-space projection, and*

$$\widehat{\boldsymbol{u}}_{t,ML}^X \triangleq (\boldsymbol{A}^* \boldsymbol{C}_t^{-1} \boldsymbol{A})^+ \boldsymbol{A}^* \boldsymbol{C}_t^{-1} \boldsymbol{u}_t^Y(\boldsymbol{y} \mid \boldsymbol{z}^Y) \tag{S8}$$

*with* $\boldsymbol{C}_t = [(a_t' \sigma_0)^2 + (b_t' \sigma_1)^2] \boldsymbol{I}_d$ *is the maximum likelihood solution of the forward model in Eq.* (10). *Note that* $\boldsymbol{A}^+$ *denotes the Moore-Penrose pseudoinverse of* $\boldsymbol{A}$, *which can be approximated by the conjugate gradient method (Appendix C.1).*

*Proof.* The proof is inspired from (Eldar, 2008). Noting the deterministic mapping between the $\mathcal{Y}$-space conditional path and the conditional vector field

$$y = \frac{a_t}{a_t'} u_t^Y(y \mid z^Y) - b_t' \left( \frac{a_t}{a_t'} - \frac{b_t}{b_t'} \right) y_1, \tag{S9}$$

we can write $v_{\boldsymbol{\theta}}^X(y, t) = v_{\boldsymbol{\theta}}^X \left( u_t^Y(y \mid z^Y), t \right)$ and the objective as

$$\mathcal{L}_{\text{PCFM}}(\boldsymbol{\theta}) = \mathbb{E}_{t, q(z^X), p_t^X(x \mid z^X), p_t^Y(y \mid z^X)} \big\| \boldsymbol{P} \big[ v_{\boldsymbol{\theta}}^X \big( u_t^Y(y \mid z^Y), t \big) - u_t^X(x \mid z^X) \big] \big\|_2^2. \tag{S10}$$

By the linear forward model over the dual-space conditional vector fields

$$u_t^Y(y \mid z^Y) = \boldsymbol{A} u_t^X(x \mid z^X) + a_t' e_0 + b_t' e_1, \tag{S11}$$

we note that the sufficient statistic $\boldsymbol{\mu}_t^X \triangleq \boldsymbol{A}^* \boldsymbol{C}_t^{-1} u_t^Y$ follows a Gaussian distribution $\mathcal{CN}(\boldsymbol{A}^* \boldsymbol{C}_t^{-1} \boldsymbol{A} u_t^X, \boldsymbol{A}^* \boldsymbol{C}_t^{-1} \boldsymbol{A})$ with probability density function (pdf)

$$p(\boldsymbol{\mu}_t^X) = q(\boldsymbol{\mu}_t^X) \exp \left( u_t^{X*} \boldsymbol{\mu}_t^X - g(u_t^X) \right), \tag{S12}$$

where

$$\begin{aligned} q(\boldsymbol{\mu}_t^X) &= K \cdot \exp \left( -\frac{1}{2} \boldsymbol{\mu}_t^{X*} \left( \boldsymbol{A}^* \boldsymbol{C}_t^{-1} \boldsymbol{A} \right)^+ \boldsymbol{\mu}_t^X \right), \\ g(u_t^X) &= \frac{1}{2} u_t^{X*} \boldsymbol{A}^* \boldsymbol{C}_t^{-1} \boldsymbol{A} u_t^X. \end{aligned} \tag{S13}$$

Assuming the network's input to be $\boldsymbol{\mu}_t^X$, we can write

$$\mathcal{L}_{\text{PCFM}}(\boldsymbol{\theta}) = \mathbb{E}_{t, q(z^X), p_t^X(x \mid z^X), p_t^Y(y \mid z^X)} \left[ u_t^{X*} \boldsymbol{P} u_t^X + v_{\boldsymbol{\theta}}^X(\boldsymbol{\mu}_t^X, t)^* \boldsymbol{P} v_{\boldsymbol{\theta}}^X(\boldsymbol{\mu}_t^X, t) - 2 u_t^{X*} \boldsymbol{P} v_{\boldsymbol{\theta}}^X(\boldsymbol{\mu}_t^X, t) \right], \tag{S14}$$

and

$$\begin{aligned} \mathbb{E}_{p_t^Y(y \mid z^X)} \left[ u_t^{X*} \boldsymbol{P} v_{\boldsymbol{\theta}}^X(\boldsymbol{\mu}_t^X, t) \right] &= \mathbb{E}_{p(\boldsymbol{\mu}_t^X)} \left[ u_t^{X*} \boldsymbol{P} v_{\boldsymbol{\theta}}^X(\boldsymbol{\mu}_t^X, t) \right] \\ &= \int v_{\boldsymbol{\theta}}^X(\boldsymbol{\mu}_t^X, t)^* \boldsymbol{P} u_t^X \cdot q(\boldsymbol{\mu}_t^X) \exp \left( u_t^{X*} \boldsymbol{\mu}_t^X - g(u_t^X) \right) \mathrm{d}\boldsymbol{\mu}_t^X. \end{aligned} \tag{S15}$$

Denote $h(\boldsymbol{\mu}_t^X) \triangleq \exp \left( u_t^{X*} \boldsymbol{\mu}_t^X - g(u_t^X) \right)$. Substituting $u_t^X h(\boldsymbol{\mu}_t^X) = \nabla_{\boldsymbol{\mu}_t^X} h(\boldsymbol{\mu}_t^X)$ and integrating by parts, we have

$$\begin{aligned} \mathbb{E}_{p(\boldsymbol{\mu}_t^X)} \left[ u_t^{X*} \boldsymbol{P} v_{\boldsymbol{\theta}}^X(\boldsymbol{\mu}_t^X, t) \right] &= \int v_{\boldsymbol{\theta}}^X(\boldsymbol{\mu}_t^X, t)^* \boldsymbol{P} u_t^X \cdot q(\boldsymbol{\mu}_t^X) \nabla_{\boldsymbol{\mu}_t^X} h(\boldsymbol{\mu}_t^X) \mathrm{d}\boldsymbol{\mu}_t^X \\ &= -\int h(\boldsymbol{\mu}_t^X) \nabla_{\boldsymbol{\mu}_t^X} \cdot \left[ q(\boldsymbol{\mu}_t^X) \boldsymbol{P} v_{\boldsymbol{\theta}}^X(\boldsymbol{\mu}_t^X, t) \right] \mathrm{d}\boldsymbol{\mu}_t^X, \end{aligned} \tag{S16}$$

where

$$\nabla_{\boldsymbol{\mu}_t^X} \cdot \left[ q(\boldsymbol{\mu}_t^X) \boldsymbol{P} v_{\boldsymbol{\theta}}^X(\boldsymbol{\mu}_t^X, t) \right] = q(\boldsymbol{\mu}_t^X) \left[ \nabla_{\boldsymbol{\mu}_t^X} \cdot \boldsymbol{P} v_{\boldsymbol{\theta}}^X(\boldsymbol{\mu}_t^X, t) + v_{\boldsymbol{\theta}}^X(\boldsymbol{\mu}_t^X, t)^* \boldsymbol{P} \nabla_{\boldsymbol{\mu}_t^X} \ln q(\boldsymbol{\mu}_t^X) \right] \tag{S17}$$

and $\ln q(\boldsymbol{\mu}_t^X) = -\left( \boldsymbol{A}^* \boldsymbol{C}_t^{-1} \boldsymbol{A} \right)^+ \boldsymbol{\mu}_t^X = -\widehat{u}_{t,\text{ML}}^X$. Therefore,

$$\mathbb{E}_{p(\boldsymbol{\mu}_t^X)} \left[ u_t^{X*} \boldsymbol{P} v_{\boldsymbol{\theta}}^X(\boldsymbol{\mu}_t^X, t) \right] = \mathbb{E}_{p(\boldsymbol{\mu}_t^X)} \left[ -\nabla_{\boldsymbol{\mu}_t^X} \cdot \boldsymbol{P} v_{\boldsymbol{\theta}}^X(\boldsymbol{\mu}_t^X, t) + v_{\boldsymbol{\theta}}^X(\boldsymbol{\mu}_t^X, t)^* \boldsymbol{P} \widehat{u}_{t,\text{ML}}^X \right] \tag{S18}$$

where

$$\begin{aligned} \mathcal{L}_{\text{PCFM}}(\boldsymbol{\theta}) &= \mathbb{E} \left[ u_t^{X*} \boldsymbol{P} u_t^X + v_{\boldsymbol{\theta}}^X(\boldsymbol{\mu}_t^X, t)^* \boldsymbol{P} v_{\boldsymbol{\theta}}^X(\boldsymbol{\mu}_t^X, t) + 2 \nabla_{\boldsymbol{\mu}_t^X} \cdot \boldsymbol{P} v_{\boldsymbol{\theta}}^X(\boldsymbol{\mu}_t^X, t) - 2 v_{\boldsymbol{\theta}}^X(\boldsymbol{\mu}_t^X, t)^* \boldsymbol{P} \widehat{u}_{t,\text{ML}}^X \right] \\ &= \mathbb{E} \left[ \big\| \boldsymbol{P} \big[ v_{\boldsymbol{\theta}}^X(\boldsymbol{\mu}_t^X, t) - \widehat{u}_{t,\text{ML}}^X \big] \big\|_2^2 + 2 \nabla_{\boldsymbol{\mu}_t^X} \cdot \boldsymbol{P} v_{\boldsymbol{\theta}}^X(\boldsymbol{\mu}_t^X, t) + \big\| \boldsymbol{P} u_t^X \big\|_2^2 - \big\| \boldsymbol{P} \widehat{u}_{t,\text{ML}}^X \big\|_2^2 \right] \\ &= \mathbb{E}_{t, q(z^X), p_t^X(x \mid z^X), p_t^Y(y \mid z^X)} \left[ \big\| \boldsymbol{P} \big[ v_{\boldsymbol{\theta}}^X(\boldsymbol{\mu}_t^X, t) - \widehat{u}_{t,\text{ML}}^X \big] \big\|_2^2 + 2 \nabla_{\boldsymbol{\mu}_t^X} \cdot \boldsymbol{P} v_{\boldsymbol{\theta}}^X(\boldsymbol{\mu}_t^X, t) \right] + \text{const.} \end{aligned} \tag{S19}$$

Then, using Eq. (S9) and writing back $\boldsymbol{v}_{\boldsymbol{\theta}}^X(\boldsymbol{\mu}_t^X, t) = \boldsymbol{v}_{\boldsymbol{\theta}}^X(\boldsymbol{A}^*\boldsymbol{y}, t)$, we obtain by change of variables that $\mathcal{L}_{\mathrm{PCFM}}(\boldsymbol{\theta})$ can be transformed to the following expression up to a constant

$$\mathbb{E}_{t, q(\boldsymbol{z}^X), p_t^Y(\boldsymbol{y}|\boldsymbol{z}^X)} \left[ \left\| \boldsymbol{P} \left[ \boldsymbol{v}_{\boldsymbol{\theta}}^X(\boldsymbol{A}^*\boldsymbol{y}, t) - \widehat{\boldsymbol{u}}_{t,\mathrm{ML}}^X \right] \right\|_2^2 + \frac{2a_t}{a_t'} [(a_t'\sigma_0)^2 + (b_t'\sigma_1)^2] \nabla_{\boldsymbol{A}^*\boldsymbol{y}} \cdot \boldsymbol{P}\boldsymbol{v}_{\boldsymbol{\theta}}^X(\boldsymbol{A}^*\boldsymbol{y}, t) \right]. \tag{S20}$$

Finally, it concludes the proof by noting that

$$\mathbb{E}_{q(\boldsymbol{z}^X)} \left[ p_t^Y(\boldsymbol{y} \mid \boldsymbol{z}^X) \right] = p_t^Y(\boldsymbol{y}) = \mathbb{E}_{q(\boldsymbol{z}^Y)} \left[ p_t^Y(\boldsymbol{y} \mid \boldsymbol{z}^Y) \right], \tag{S21}$$

where $q(\boldsymbol{z}^Y) = q(\boldsymbol{y}_0)\mathbb{E}_{q(\boldsymbol{x}_1)} \left[ p(\boldsymbol{y}_1 \mid \boldsymbol{x}_1) \right]$ is sampled by the MRI and Monte Carlo estimation. $\square$

**Lemma 1.** *The $\mathcal{Y}$-space marginal vector field that generates the probability path $p_t^Y$ takes the form*

$$\boldsymbol{u}_t^Y(\boldsymbol{y}) = \boldsymbol{A}\boldsymbol{v}_{\boldsymbol{\theta}^*}^X(\boldsymbol{y}, t) - (a_t a_t'\sigma_0^2 + b_t b_t'\sigma_1^2)\nabla_{\boldsymbol{y}} \log p_t^Y(\boldsymbol{y}). \tag{S22}$$

*Proof.* Eq. (16) shows that the $\mathcal{Y}$-space conditional vector field is

$$\boldsymbol{u}_t^Y(\boldsymbol{y} \mid \boldsymbol{z}^X) = \boldsymbol{A}\boldsymbol{u}_t^X(\boldsymbol{x} \mid \boldsymbol{z}^X) - (a_t a_t'\sigma_0^2 + b_t b_t'\sigma_1^2)\nabla_{\boldsymbol{y}} \log p_t^Y(\boldsymbol{y} \mid \boldsymbol{z}^X). \tag{S23}$$

Therefore, by Proposition 1, the $\mathcal{Y}$-space marginal vector field takes the form

$$\begin{aligned}
\boldsymbol{u}_t^Y(\boldsymbol{y}) &= \mathbb{E}_{q_t(\boldsymbol{z}^X|\boldsymbol{y})} \left[ \boldsymbol{u}_t^Y(\boldsymbol{y} \mid \boldsymbol{z}^X) \right] \\
&= \boldsymbol{A}\mathbb{E}_{q_t(\boldsymbol{z}^X|\boldsymbol{y})} \left[ \boldsymbol{u}_t^X(\boldsymbol{x} \mid \boldsymbol{z}^X) \right] - (a_t a_t'\sigma_0^2 + b_t b_t'\sigma_1^2)\mathbb{E}_{q_t(\boldsymbol{z}^X|\boldsymbol{y})} \left[ \nabla_{\boldsymbol{y}} \log p_t^Y(\boldsymbol{y} \mid \boldsymbol{z}^X) \right] \\
&= \boldsymbol{A}\boldsymbol{v}_{\boldsymbol{\theta}^*}^X(\boldsymbol{y}, t) - (a_t a_t'\sigma_0^2 + b_t b_t'\sigma_1^2)\nabla_{\boldsymbol{y}} \log p_t^Y(\boldsymbol{y}),
\end{aligned} \tag{S24}$$

where we have used the fact that

$$\begin{aligned}
\mathbb{E}_{q_t(\boldsymbol{z}^X|\boldsymbol{y})} \left[ \nabla_{\boldsymbol{y}} \log p_t^Y(\boldsymbol{y} \mid \boldsymbol{z}^X) \right] &= \int q_t(\boldsymbol{z}^X \mid \boldsymbol{y}) \frac{1}{p_t^Y(\boldsymbol{y} \mid \boldsymbol{z}^X)} \nabla_{\boldsymbol{y}} p_t^Y(\boldsymbol{y} \mid \boldsymbol{z}^X) \mathrm{d}\boldsymbol{z}^X \\
&= \frac{1}{p_t^Y(\boldsymbol{y})} \int q(\boldsymbol{z}^X) \nabla_{\boldsymbol{y}} p_t^Y(\boldsymbol{y} \mid \boldsymbol{z}^X) \mathrm{d}\boldsymbol{z}^X \\
&= \frac{1}{p_t^Y(\boldsymbol{y})} \nabla_{\boldsymbol{y}} \int q(\boldsymbol{z}^X) p_t^Y(\boldsymbol{y} \mid \boldsymbol{z}^X) \mathrm{d}\boldsymbol{z}^X \\
&= \frac{1}{p_t^Y(\boldsymbol{y})} \nabla_{\boldsymbol{y}} p_t^Y(\boldsymbol{y}) \\
&= \nabla_{\boldsymbol{y}} \log p_t^Y(\boldsymbol{y}).
\end{aligned} \tag{S25}$$

$\square$

**Lemma 2.** *Note that $p_1^Y(\boldsymbol{y}) = \int p_1(\boldsymbol{y} \mid \boldsymbol{x}) p_1^X(\boldsymbol{x}) \mathrm{d}\boldsymbol{x} = \mathcal{CN}(\boldsymbol{y} \mid \boldsymbol{0}, 2\boldsymbol{A}\boldsymbol{A}^* + \sigma_1^2 \boldsymbol{I}_{Cd})$. Then,*

$$\boldsymbol{u}_t^Y(\boldsymbol{y}) = \frac{a_t'}{a_t}\boldsymbol{y} - b_t\left(b_t' - \frac{a_t'}{a_t}b_t\right)(2\boldsymbol{A}\boldsymbol{A}^* + \sigma_1^2 \boldsymbol{I}_{Cd})\nabla_{\boldsymbol{y}} \log p_t^Y(\boldsymbol{y}). \tag{S26}$$

*Proof.* Taking $\boldsymbol{y}_0$ as the conditioning variable, we have

$$\begin{aligned}
\boldsymbol{u}_t^Y(\boldsymbol{y}) &= \mathbb{E}_{q_t(\boldsymbol{y}_0|\boldsymbol{y})} \left[ a_t'\boldsymbol{y}_0 + b_t'\boldsymbol{y}_1 \right] \\
&= \mathbb{E}_{q_t(\boldsymbol{y}_0|\boldsymbol{y})} \left[ a_t' \frac{\boldsymbol{y} - b_t \boldsymbol{y}_1}{a_t} + b_t'\boldsymbol{y}_1 \right] \\
&= \frac{a_t'}{a_t}\boldsymbol{y} + \left( b_t' - \frac{a_t'}{a_t}b_t \right) \mathbb{E}_{q_t(\boldsymbol{y}_0|\boldsymbol{y})}[\boldsymbol{y}_1],
\end{aligned} \tag{S27}$$

where $\boldsymbol{y}_1 = \frac{\boldsymbol{y} - a_t \boldsymbol{y}_0}{b_t}$.

On the other hand, noting that $p_t^Y(\boldsymbol{y} \mid \boldsymbol{y}_0) = \mathcal{CN}\left(\boldsymbol{y} \mid a_t\boldsymbol{y}_0, b_t^2(2\boldsymbol{A}\boldsymbol{A}^* + \sigma_1^2\boldsymbol{I}_{Cd})\right)$, the score function can be written as

$$
\begin{aligned}
\nabla_{\boldsymbol{y}} \log p_t^Y(\boldsymbol{y}) &= \frac{1}{p_t^Y(\boldsymbol{y})} \nabla_{\boldsymbol{y}} p_t^Y(\boldsymbol{y}) \\
&= \frac{1}{p_t^Y(\boldsymbol{y})} \nabla_{\boldsymbol{y}} \int p_t^Y(\boldsymbol{y} \mid \boldsymbol{y}_0) q(\boldsymbol{y}_0) \mathrm{d}\boldsymbol{y}_0 \\
&= \frac{1}{p_t^Y(\boldsymbol{y})} \int p_t^Y(\boldsymbol{y} \mid \boldsymbol{y}_0) q(\boldsymbol{y}_0) \nabla_{\boldsymbol{y}} \log p_t^Y(\boldsymbol{y} \mid \boldsymbol{y}_0) \mathrm{d}\boldsymbol{y}_0 \\
&= -\frac{1}{p_t^Y(\boldsymbol{y})} \int p_t^Y(\boldsymbol{y} \mid \boldsymbol{y}_0) q(\boldsymbol{y}_0) \frac{(2\boldsymbol{A}\boldsymbol{A}^* + \sigma_1^2\boldsymbol{I}_{Cd})^{-1}(\boldsymbol{y} - a_t\boldsymbol{y}_0)}{b_t^2} \mathrm{d}\boldsymbol{y}_0 \\
&= -\int q_t(\boldsymbol{y}_0 \mid \boldsymbol{y}) \frac{(2\boldsymbol{A}\boldsymbol{A}^* + \sigma_1^2\boldsymbol{I}_{Cd})^{-1} b_t\boldsymbol{y}_1}{b_t^2} \mathrm{d}\boldsymbol{y}_0 \\
&= -\frac{(2\boldsymbol{A}\boldsymbol{A}^* + \sigma_1^2\boldsymbol{I}_{Cd})^{-1}}{b_t} \mathbb{E}_{q_t(\boldsymbol{y}_0 \mid \boldsymbol{y})}[\boldsymbol{y}_1].
\end{aligned}
\tag{S28}
$$

Combining Eq. (S27) and Eq. (S28) by canceling out $\mathbb{E}_{q_t(\boldsymbol{y}_0 \mid \boldsymbol{y})}[\boldsymbol{y}_1]$ completes the proof. $\qquad\square$

**Proposition 3 (Vector fields under projection).** *For $a_t = 1 - t$ and $b_t = t$, the $\mathcal{Y}$-space marginal vector field $\boldsymbol{u}_t^Y(\boldsymbol{y})$ can be expressed by $\boldsymbol{v}_{\boldsymbol{\theta}^*}^X(\boldsymbol{y}, t)$ as*

$$
\boldsymbol{u}_t^Y(\boldsymbol{y}) = \boldsymbol{A}\boldsymbol{v}_{\boldsymbol{\theta}^*}^X(\boldsymbol{y}, t) - \frac{c_t}{1-t}\left[(c_t + \sigma_1^2)\boldsymbol{I}_{Cd} + 2\boldsymbol{A}\boldsymbol{A}^*\right]^{-1}\left[(1-t)\boldsymbol{A}\boldsymbol{v}_{\boldsymbol{\theta}^*}^X(\boldsymbol{y}, t) + \boldsymbol{y}\right], \tag{S29}
$$

*where $c_t \triangleq (1 - t)\left(\frac{1-t}{t}\sigma_0^2 - \sigma_1^2\right)$. In addition, left-multiplying both sides with $\boldsymbol{A}^*$ gives the more computationally friendly formula when $Cd > D$:*

$$
\boldsymbol{A}^*\boldsymbol{u}_t^Y(\boldsymbol{y}) = \boldsymbol{A}^*\boldsymbol{A}\boldsymbol{v}_{\boldsymbol{\theta}^*}^X(\boldsymbol{y}, t) - \frac{c_t}{1-t}\left[(c_t + \sigma_1^2)\boldsymbol{I}_D + 2\boldsymbol{A}^*\boldsymbol{A}\right]^{-1}\boldsymbol{A}^*\left[(1-t)\boldsymbol{A}\boldsymbol{v}_{\boldsymbol{\theta}^*}^X(\boldsymbol{y}, t) + \boldsymbol{y}\right].
\tag{S30}
$$

*Proof.* For $a_t = 1 - t$ and $b_t = t$, Lemma 2 indicates

$$
\nabla_{\boldsymbol{y}} \log p_t^Y(\boldsymbol{y}) = -\frac{1}{t}(2\boldsymbol{A}\boldsymbol{A}^* + \sigma_1^2\boldsymbol{I}_{Cd})^{-1}\left[\boldsymbol{y} + (1-t)\boldsymbol{u}_t^Y(\boldsymbol{y})\right]. \tag{S31}
$$

Substituting this into Lemma 1 gives the equation

$$
\boldsymbol{u}_t^Y(\boldsymbol{y}) = \boldsymbol{A}\boldsymbol{v}_{\boldsymbol{\theta}^*}^X(\boldsymbol{y}, t) - \left(\frac{1-t}{t}\sigma_0^2 - \sigma_1^2\right)(2\boldsymbol{A}\boldsymbol{A}^* + \sigma_1^2\boldsymbol{I}_{Cd})^{-1}\left[\boldsymbol{y} + (1-t)\boldsymbol{u}_t^Y(\boldsymbol{y})\right]. \tag{S32}
$$

Denoting $\boldsymbol{u} \triangleq \boldsymbol{u}_t^Y(\boldsymbol{y})$ and $\boldsymbol{v} \triangleq \boldsymbol{v}_{\boldsymbol{\theta}^*}^X(\boldsymbol{y}, t)$, then solving for $\boldsymbol{u}$ in the above equation gives

$$
\boldsymbol{u} = \left(\boldsymbol{I} + c(2\boldsymbol{A}\boldsymbol{A}^* + \sigma_1^2\boldsymbol{I}_{Cd})^{-1}\right)^{-1}\left(\boldsymbol{A}\boldsymbol{v} - \frac{c}{1-t}(2\boldsymbol{A}\boldsymbol{A}^* + \sigma_1^2\boldsymbol{I}_{Cd})^{-1}\boldsymbol{y}\right), \tag{S33}
$$

where $c \triangleq \left(\frac{1-t}{t}\sigma_0^2 - \sigma_1^2\right)(1 - t)$.

By the Woodbury matrix identity $(\boldsymbol{A} + \boldsymbol{U}\boldsymbol{C}\boldsymbol{V})^{-1} = \boldsymbol{A}^{-1} - \boldsymbol{A}^{-1}\boldsymbol{U}(\boldsymbol{C}^{-1} + \boldsymbol{V}\boldsymbol{A}^{-1}\boldsymbol{U}^{-1})^{-1}\boldsymbol{V}\boldsymbol{A}^{-1}$, we have

$$
\left(\boldsymbol{I} + c(2\boldsymbol{A}\boldsymbol{A}^* + \sigma_1^2\boldsymbol{I})^{-1}\right)^{-1} = \boldsymbol{I} - c\left(c\boldsymbol{I} + 2\boldsymbol{A}\boldsymbol{A}^* + \sigma_1^2\boldsymbol{I}\right)^{-1}, \tag{S34}
$$

and note that

$$
\left(\boldsymbol{I} + c(2\boldsymbol{A}\boldsymbol{A}^* + \sigma_1^2\boldsymbol{I})^{-1})\right)^{-1}(2\boldsymbol{A}\boldsymbol{A}^* + \sigma_1^2\boldsymbol{I})^{-1} = \left(c\boldsymbol{I} + 2\boldsymbol{A}\boldsymbol{A}^* + \sigma_1^2\boldsymbol{I}\right)^{-1}. \tag{S35}
$$

Therefore,

$$
\begin{aligned}
\boldsymbol{u} &= \left[\boldsymbol{I} - c\left(c\boldsymbol{I} + 2\boldsymbol{A}\boldsymbol{A}^* + \sigma_1^2\boldsymbol{I}\right)^{-1}\right]\boldsymbol{A}\boldsymbol{v} - \frac{c}{1-t}\left(c\boldsymbol{I} + 2\boldsymbol{A}\boldsymbol{A}^* + \sigma_1^2\boldsymbol{I}\right)^{-1}\boldsymbol{y} \\
&= \boldsymbol{A}\boldsymbol{v} - c\left(c\boldsymbol{I} + 2\boldsymbol{A}\boldsymbol{A}^* + \sigma_1^2\boldsymbol{I}\right)^{-1}\left[\boldsymbol{A}\boldsymbol{v} + \frac{1}{1-t}\boldsymbol{y}\right] \\
&= \boldsymbol{A}\boldsymbol{v} - \frac{c}{1-t}\left(c\boldsymbol{I} + 2\boldsymbol{A}\boldsymbol{A}^* + \sigma_1^2\boldsymbol{I}\right)^{-1}\left[(1-t)\boldsymbol{A}\boldsymbol{v} + \boldsymbol{y}\right],
\end{aligned}
\tag{S36}
$$

and by the identity $\boldsymbol{A}^*(c\boldsymbol{I}_{Cd} + 2\boldsymbol{A}\boldsymbol{A}^* + \sigma_1^2\boldsymbol{I}_{Cd})^{-1} = (c\boldsymbol{I}_D + 2\boldsymbol{A}^*\boldsymbol{A} + \sigma_1^2\boldsymbol{I}_D)^{-1}\boldsymbol{A}^*$,

$$
\boldsymbol{A}^*\boldsymbol{u} = \boldsymbol{A}^*\boldsymbol{A}\boldsymbol{v} - \frac{c}{1-t}(c\boldsymbol{I}_D + 2\boldsymbol{A}^*\boldsymbol{A} + \sigma_1^2\boldsymbol{I})^{-1}\boldsymbol{A}^*\left[(1-t)\boldsymbol{A}\boldsymbol{v} + \boldsymbol{y}\right]. \tag{S37}
$$

$\qquad\square$

---

**Algorithm 2:** PnP Cyclic Integration with PCFM for Noisy Measurements

---

**Input:** k-space measurement $\boldsymbol{y}_0$, pretrained optimal solution to PCFM $\boldsymbol{v}_{\boldsymbol{\theta}^*}^X(\cdot, t)$, number of time steps $T$, adaptive step size $\gamma_t$.

**Output:** Reconstructed image $\boldsymbol{x}_0$ of $\boldsymbol{y}_0$.

**1 for** $t = 0, \ldots, {}^{(T-1)}/T$ **do**

**2**    $\boldsymbol{y}_{t+1/T} \leftarrow \boldsymbol{y}_t + \frac{1}{T}\boldsymbol{u}_t^Y(\boldsymbol{y})$             ▷ Forward integration using Proposition 3

**3 Sample** $\boldsymbol{x}_1 \sim p_1(\boldsymbol{x} \mid \boldsymbol{y})$.             ▷ Posterior sampling with Eq. (22)

**4 for** $t \in \{1, \ldots, {}^1/T\}$ **do**

**5**    $\widetilde{\boldsymbol{y}}_t \leftarrow a_t \boldsymbol{y}_0 + b_t \boldsymbol{y}_1$

**6**    $\widetilde{\boldsymbol{x}}_0 \leftarrow \boldsymbol{x}_t - t\boldsymbol{v}_{\boldsymbol{\theta}^*}^X(\widetilde{\boldsymbol{y}}_t, t)$             ▷ PnP-Flow denoising step

**7**    $\boldsymbol{x}_0 \leftarrow \widetilde{\boldsymbol{x}}_0 - \gamma_t \boldsymbol{A}^*(\boldsymbol{A}\widetilde{\boldsymbol{x}}_0 - \widetilde{\boldsymbol{y}}_0)$             ▷ Gradient step

**8**    $\boldsymbol{x}_{t-1/T} \leftarrow a_{t-1/T}\boldsymbol{x}_0 + b_{t-1/T}\boldsymbol{x}_1$             ▷ Interpolation step

**9 return** $\boldsymbol{x}_0$.

---

## B    PLUG-AND-PLAY CYCLIC INTEGRATION WITH PCFM FOR NOISY MEASUREMENTS

The plug-and-play (PnP) framework (Venkatakrishnan et al., 2013; Fang et al., 2024) utilizes off-the-shelf denoising techniques to address the general inverse problem while optimizing the objective function

$$\min_{\boldsymbol{x}} \left\{ \frac{1}{2} \|\boldsymbol{y} - \mathcal{A}(\boldsymbol{x})\|_2^2 + R(\boldsymbol{x}) \right\}, \tag{S38}$$

where $R(\boldsymbol{x})$ serves as a regularizer, encouraging solutions that are probable under the prior distribution of $\boldsymbol{x}$. PnP substitutes the explicit solution of the proximal operator

$$\mathrm{prox}_R(\boldsymbol{y}) \triangleq \arg\min_{\boldsymbol{x}} \left\{ \frac{1}{2} \|\boldsymbol{y} - \boldsymbol{x}\|_2^2 + R(\boldsymbol{x}) \right\}, \tag{S39}$$

which is utilized to optimize the objective through methods such as proximal gradient descent (PGD) (Beck, 2017), half quadratic splitting (HQS) (Geman & Yang, 1995), and alternating direction methods of multipliers (ADMM) (Boyd et al., 2011). In situations involving noisy k-space data, we utilize the PGD and PnP-Flow frameworks (Martin et al., 2025). These frameworks alternate between a denoising step using a pretrained flow matching model, aiming to approximate the proximal operator's solution, and a gradient step to promote data consistency. Alg. 2 presents the discrete-time algorithm implementing the PnP-based cyclic integration with the proposed PCFM framework.

## C    NUMERICAL METHODS

### C.1    CONJUGATE GRADIENT

Conjugate gradient (CG) is a method for solving a linear system of equations $\boldsymbol{A}\boldsymbol{x} = \boldsymbol{b}$, when the matrix $\boldsymbol{A}$ is symmetric (or Hermitian) positive-definite (SPD) and very large (often sparse). Direct methods like Gaussian elimination are impractical due to computational cost and memory requirements. CG reframes the problem of solving a linear system as an optimization problem. The solution to $\boldsymbol{A}\boldsymbol{x} = \boldsymbol{b}$ is precisely the vector $\boldsymbol{x}$ that minimizes the quadratic form:

$$\phi(\boldsymbol{x}) = \frac{1}{2}\boldsymbol{x}^* \boldsymbol{A}\boldsymbol{x} - \boldsymbol{x}^*\boldsymbol{b}. \tag{S40}$$

An intuitive way to find this minimum is to take the steepest descent, where one repeatedly steps in the direction of the negative gradient $-\nabla\phi(\boldsymbol{x}) = \boldsymbol{b} - \boldsymbol{A}\boldsymbol{x}$. However, the steepest descent often performs poorly, taking many small zigzagging steps to reach the minimum.

CG dramatically improves upon this by choosing a sequence of search directions that are "smarter". Instead of using the residual at each step, it generates a set of search directions $\{\boldsymbol{p}_0, \boldsymbol{p}_1, \ldots, \boldsymbol{p}_{k-1}\}$ that are mutually $\boldsymbol{A}$-orthogonal, i.e., $\boldsymbol{p}_i^* \boldsymbol{A}\boldsymbol{p}_j = 0$ for $i \neq j$. This property is crucial: minimizing the

---

**Algorithm 3:** Conjugate Gradient (CG)

---

**Input:** A symmetric (or Hermitian) matrix $\boldsymbol{A}$, a vector $\boldsymbol{b}$, an intial guess $\boldsymbol{x}_0$, a maximum number of iterations $k$, and a tolerance $\epsilon$.

**Output:** An approximate solution $\boldsymbol{x}$ to $\boldsymbol{Ax} = \boldsymbol{b}$.

1 Set $\boldsymbol{r}_0 \leftarrow \boldsymbol{b} - \boldsymbol{Ax}_0$ and $\boldsymbol{p}_0 \leftarrow \boldsymbol{r}_0$. $\qquad\qquad$ ▷ Initialization

2 **for** $j = 1, \ldots, k$ **do**

3 $\quad$ $\boldsymbol{v}_j \leftarrow \boldsymbol{Ap}_j$

4 $\quad$ $\alpha_j \leftarrow \dfrac{\boldsymbol{r}_j^* \boldsymbol{r}_j}{\boldsymbol{p}_j^* \boldsymbol{v}_j}$ $\qquad\qquad$ ▷ Compute the step size

5 $\quad$ $\boldsymbol{x}_{j+1} \leftarrow \boldsymbol{x}_j + \alpha_j \boldsymbol{p}_j$ $\qquad\qquad$ ▷ Update solution

6 $\quad$ $\boldsymbol{r}_{j+1} \leftarrow \boldsymbol{r}_j - \alpha_j \boldsymbol{v}_j$ $\qquad\qquad$ ▷ Update residual

7 $\quad$ **if** $\|\boldsymbol{r}_{j+1}\|_2 < \epsilon$ **then**

8 $\qquad$ **break**

9 $\quad$ $\beta_j \leftarrow \dfrac{\boldsymbol{r}_{j+1}^* \boldsymbol{r}_{j+1}}{\boldsymbol{r}_j^* \boldsymbol{r}_j}$ $\qquad\qquad$ ▷ Calculate the improvement factor

10 $\quad$ $\boldsymbol{p}_{j+1} \leftarrow \boldsymbol{r}_{j+1} + \beta_j \boldsymbol{p}_j$ $\qquad\qquad$ ▷ Update search direction

11 **return** $\boldsymbol{x}_{j+1}$.

---

quadratic function along a new search direction $\boldsymbol{p}_k$ does not compromise the minimization that has already been achieved in the previous directions. At each iteration $k$, CG finds the optimal solution $\boldsymbol{x}_k$ within the affine Krylov subspace $\boldsymbol{x}_0 + \mathcal{K}_k(\boldsymbol{A}, \boldsymbol{r}_0)$, where $\boldsymbol{r}_0 \triangleq \boldsymbol{b} - \boldsymbol{Ax}_0$ is the initial residual. This means that after $k$ steps, CG has found the best possible solution that can be formed by a linear combination of $\{\boldsymbol{r}_0, \boldsymbol{Ar}_0, \ldots, \boldsymbol{A}^{k-1}\boldsymbol{r}_0\}$. This guarantees convergence for an $N \times N$ matrix in at most $N$ steps, though in practice, a good approximation is often found in far fewer iterations. Alg. 3 provides the algorithm in detail.

## D DETAILS OF THE COMPARED BASELINES

We benchmark our method against three types of baseline approaches.

(A) Supervised methods that require fully sampled MRIs during training:

- **MoDL** (Aggarwal et al., 2018) is a model-based end-to-end network that unrolls traditional optimization procedure by regarding the CNN as a regularizer. We use 10 unrolling iterations of the network. Training is performed by minimizing the L2 loss between the network output and the ground truth.
- **DDNM⁺** (Wang et al., 2023) is a diffusion model-based method for inverse problems solving. Training is performed by denoising on fully sampled images with the DDPM framework (Ho et al., 2020), whereas the inference is implemented by alternating between the DDIM sampling steps (Song et al., 2021) and the range-null decomposition for enforcing measurement consistency. Training is performed by the noise prediction objective with the cosine noise schedule of iDDPM (Nichol & Dhariwal, 2021). We use the same network architecture and training strategy as our proposed approach for this method.
- **OT-ODE** (Pokle et al., 2024) estimates the posterior vector field by combining the original vector field learned from fully sampled images with the likelihood score approximated by the ΠGDM estimation (Song et al., 2023). Training is performed by optimizing the original CFM objective (Lipman et al., 2023). We use the same network architecture and training strategy as our proposed approach for this method.
- **PnP-Flow** (Martin et al., 2025) integrates the Plug-and-Play framework with flow matching, which alternates between gradient descent steps for measurement consistency, reprojections onto the learned flow path, and denoising by the pre-trained vector field. Training is performed by optimizing the original CFM objective (Lipman et al., 2023). We use the same network architecture and training strategy as our proposed approach for this method. We set the hyperparameter $\gamma_t = t^\alpha$ with $\alpha = 0.1$.

(B) Self-supervised methods that learn to reconstruct by additional subsampling of the available k-space measurements.

- **SSDU** (Yaman et al., 2020) proposes to split the available k-space measurements into two disjoint subsets and then train a model-based reconstruction network to recover one of the subsets from the other. We use the VarNet (Hammernik et al., 2018) with 10 unrolling iterations as the network backbone. Training is achieved by minimizing the L2 loss.

- **Weighted SSDU** (Millard & Chiew, 2023) improves upon the SSDU framework by using a subsampling mask of the same distribution as the original mask and re-weighting the L2 loss. We use the VarNet (Hammernik et al., 2018) with 10 unrolling iterations as the network backbone.

- **Robust SSDU** (Millard & Chiew, 2024) provably recovers clean images from noisy, undersampled training data by simultaneously estimating missing k-space samples and denoising the available samples. We use the VarNet (Hammernik et al., 2018) with 10 unrolling iterations as the network backbone.

(C) Unsupervised methods that learn to reconstruct using all the available k-space measurements.

- **REI** (Chen et al., 2022) achieves unsupervised reconstruction by combining the k-space SURE-based loss (Stein, 1981) for measurement consistency and the equivariant imaging framework which builds on the group invariance assumption of the signal space (Chen et al., 2021; Tachella et al., 2023). We use the VarNet (Hammernik et al., 2018) with 10 unrolling iterations as the network backbone.

- **MOI** (Tachella et al., 2022) leverages the randomness in the imaging operator and proposes an unsupervised loss that ensures consistency across all operators. We use the MoDL (Aggarwal et al., 2018) architecture with 10 unrolling iterations as the network backbone.

- **ENSURE** (Aggarwal et al., 2022) also leverages the randomness in the forward operators and provides an unbiased estimate of the true mean squared error without fully sampled images. We use the MoDL (Aggarwal et al., 2018) architecture with 10 unrolling iterations as the network backbone. However, it uses an inaccurate numerical strategy to approximate the loss function in the multi-coil scenario.

- **GTF$^2$M** (Luo et al., 2025) proposes a ground-truth-free flow matching framework for single-coil MRI. Reconstruction for multi-coil MRI is achieved by coil-wise reconstruction followed by SENSE-based combination. We use the same network architecture and training strategy as our proposed approach for this method. Nevertheless, the performance of this method is suboptimal as the prior is learned from single-coil k-space measurements instead of the combined forward operator of parallel MRI.

# E  ADDITIONAL RESULTS

## E.1  INFERENCE TIME

Table S1 illustrates the average inference time of the various methods evaluated for the reconstruction of a single image from the CMRxRecon 2023 dataset. It is evident that among the generative model-based techniques, our method demonstrates the fastest inference time.

## E.2  ABLATION STUDY ON BACKWARD SAMPLING

Table S2 displays the results of the ablation study examining various backward sampling strategies applied to the original initial noiseless CMRxRecon 2023 dataset. It can be noted that the PnP-based backward sampling technique is less effective compared to Alg. 1 when tested on noiseless data.

## E.3  ABLATION STUDY ON FORWARD SAMPLING

If the forward integration is omitted, the unconditional distribution $p(\boldsymbol{y}_1 \mid \boldsymbol{y}_0)$ will be assumed to be $\mathcal{CN}(\boldsymbol{0}, \boldsymbol{I}_d)$. Table S3 displays results from the ablation study investigating the impact of incorporating the proposed forward integration steps. It is evident that incorporating forward integration steps markedly enhances reconstruction performance.

Table S1: Average inference time of the compared methods for reconstructing one image from the CMRxRecon 2023 dataset. The inference time is calculated on an NVIDIA A5000 GPU using a batch size of 4.

| Training Supervision | Reconstruction Method | Time (ms) | NFEs ↓ |
|---|---|---|---|
| Supervised | MoDL | 58 | 1 |
| | DDNM⁺ | 1000 | 100 |
| | OT-ODE | 1750 | 100 |
| | PnP-Flow | 938 | 100 |
| Self-supervised | SSDU | 19 | 1 |
| | Weighted SSDU | 19 | 1 |
| | Robust SSDU | 19 | 1 |
| Unsupervised | REI | 19 | 1 |
| | MOI | 58 | 1 |
| | ENSURE | 19 | 1 |
| | GTF$^2$M | 1563 | 20 |
| | PCFM (Ours) | 500 | 20 |

Table S2: Ablation study on the backward integration strategy on the CMRxRecon 2023 cardiac T1/T2 mapping MRI (noiseless). The difference in metrics is statistically significant between the two strategies by the two-sided paired $t$-test ($p < 0.05$).

| Reconstruction Method | SSIM ↑ | | PSNR ↑ | | NFEs ↓ |
|---|---|---|---|---|---|
| | 4× | 8× | 4× | 8× | |
| Zero-Filled | $0.769 \pm 0.057$ | $0.747 \pm 0.056$ | $27.01 \pm 1.97$ | $26.11 \pm 1.90$ | N/A |
| PnP-PCFM | $0.808 \pm 0.057$ | $0.893 \pm 0.038$ | $28.82 \pm 2.19$ | $33.16 \pm 2.82$ | 20 |
| PCFM | $0.994 \pm 0.008$ | $0.974 \pm 0.016$ | $52.39 \pm 9.81$ | $40.50 \pm 3.84$ | 20 |

### E.4 NOISY DATA

Table S4 displays the quantitative outcomes on the noise-free test data from the CMRxRecon 2023 dataset. A notable observation is that the model retains its performance as though the training data were free of noise, suggesting that PCFM effectively learns the prior distribution of fully sampled MRI even when trained on noisy and undersampled inputs.

Fig. S2 illustrates the reconstruction outcomes from cardiac MRI when subjected to additive Gaussian noise at different levels. It can be noted that with an increase in noise level, the reconstruction is more susceptible to contamination by measurement noise.

Table S3: Ablation study on the forward sampling strategy on the CMRxRecon 2023 cardiac T1/T2 mapping MRI (noiseless). The difference in metrics is statistically significant between the two strategies by the two-sided paired $t$-test ($p < 0.05$).

| Reconstruction Method | SSIM ↑ | | PSNR ↑ | | NFEs ↓ |
|---|---|---|---|---|---|
| | 4× | 8× | 4× | 8× | |
| Zero-Filled | $0.769 \pm 0.057$ | $0.747 \pm 0.056$ | $27.01 \pm 1.97$ | $26.11 \pm 1.90$ | N/A |
| PCFM w/o forward | $0.988 \pm 0.012$ | $0.960 \pm 0.021$ | $46.21 \pm 6.00$ | $38.12 \pm 3.51$ | 10 |
| PCFM | $0.994 \pm 0.008$ | $0.974 \pm 0.016$ | $52.39 \pm 9.81$ | $40.50 \pm 3.84$ | 20 |

Table S4: Qualitative results of $4\times$ and $8\times$ parallel MRI reconstruction of CMRxRecon 2023 cardiac T1/T2 quantitative mapping images using PCFM trained on noisy data while tested on clean data.

| Training Noise level | Reconstruction Method | SSIM ↑ | | PSNR ↑ | | NFEs ↓ |
|---|---|---|---|---|---|---|
| | | $4\times$ | $8\times$ | $4\times$ | $8\times$ | |
| $\sigma_0 = 0.05$ | PCFM (Ours) | $0.995 \pm 0.005$ | $0.974 \pm 0.016$ | $51.34 \pm 6.21$ | $40.51 \pm 3.84$ | 20 |
| $\sigma_0 = 0.1$ | PCFM (Ours) | $0.995 \pm 0.005$ | $0.974 \pm 0.016$ | $51.34 \pm 6.19$ | $40.50 \pm 3.83$ | 20 |
| $\sigma_0 = 0.2$ | PCFM (Ours) | $0.995 \pm 0.005$ | $0.974 \pm 0.016$ | $51.30 \pm 6.21$ | $40.55 \pm 3.83$ | 20 |

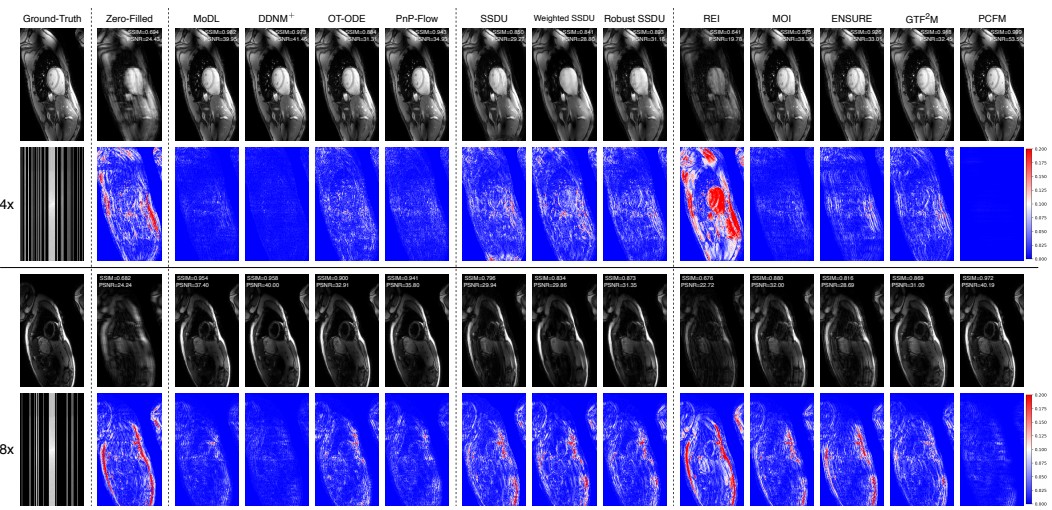

Figure S1: Visualization of reconstruction on two test samples of $4\times$ and $8\times$ accelerated multi-coil cardiac MRI from the compared methods. The k-space are presented in log-scale absolute values. The error maps are presented in values relative to the peak intensity in the ground-truth image.

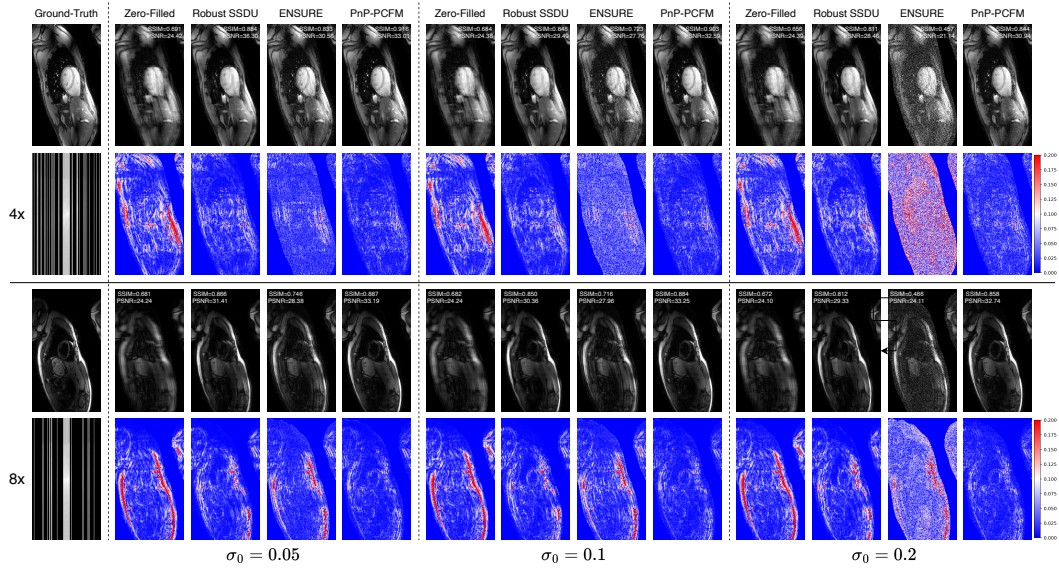

Figure S2: Visualization of reconstruction on two test samples of $4\times$ and $8\times$ accelerated multi-coil cardiac MRI with additive Gaussian noise of various scales.

