# OpenReview forum: "Unsupervised Parallel MRI Reconstruction via Projected Conditional Flow Matching"
_ICLR.cc/2026/Conference — ICLR 2026 Conference Withdrawn Submission_

### Official Review · Reviewer_gf2W · 2025-10-31

**Soundness:** 3
**Presentation:** 2
**Contribution:** 1
**Rating:** 2
**Confidence:** 5

**Summary:**

The paper proposes an unsupervised framework for parallel MRI reconstruction, termed Projected Conditional Flow Matching (PCFM). The method aims to learn a denoising process of fully sampled MR images using only undersampled $k$-space measurements, without requiring ground-truth data. Specifically, the authors formulate a PCFM objective inspired by the Generalized Stein's Unbiased Risk Estimator (GSURE). This allows learning the signal-space vector field through the MRI forward operator as a projection. A dual-space cyclic reconstruction algorithm is further introduced, performing forward and backward ODE integrations in both image and measurement spaces with data-consistency projections. Experiments with fastMRI and CMRxRecon show that PCFM achieves comparable or superior PSNR/SSIM to supervised and unsupervised baselines, while being computationally efficient (approximately 20 NFEs).

**Strengths:**

1. The paper addresses an important and practically challenging problem: MRI reconstruction without access to fully sampled ground-truth data.

2. The derivation connecting signal- and measurement-space conditional vector fields via $\mathbf{u}_t^Y = \mathbf{A} \mathbf{u}_t^X + a'_t \mathbf{e}_0 + b'_t \mathbf{e}_1$ is consistent and well-motivated.
The GSURE-inspired objective provides a theoretically clean way to train without paired supervision.

3. The cyclic dual-space integration offers a balance between the interpretability of unrolled optimization and the flexibility of flow-based priors, achieving efficient inference.

4. On the fastMRI knee and CMRxRecon datasets, PCFM performs on par with supervised methods and significantly outperforms previous unsupervised ones, validating the framework.

5. The manuscript includes detailed derivations.

**Weaknesses:**

1. **Substantial overlap with prior accepted work (IPMI 2025):**  The core technical content---including the formulation of dual-space conditional vector fields, the ground-truth-free flow-matching objective, the unbiased risk estimator (ENSURE/GSURE style), and the cyclic forward--backward reconstruction algorithm---is virtually identical to the paper "Unsupervised Accelerated MRI Reconstruction via Ground-Truth-Free Flow Matching (GTF$^2$M)" by Luo et al., already accepted to IPMI 2025.
That IPMI paper introduces a nearly identical induced forward model $\mathbf{u}_t^Y = \mathbf{A} \mathbf{u}_t^X + a'_t \mathbf{e}$ and defines a very similar loss: $L\_{\text{GTF$^2$M}} = \mathbb{E} \|| \mathbf{R}_s  \[ \mathbf{h}\_\theta (\mathbf{A}^\ast \mathbf{z}_t^Y, t) - \hat{\mathbf{u}}\_{t,s}^{X,\text{ML}} \] \||_2^2 + 2 a'_t a_t \sigma^2 \nabla\_{\mathbf{A}^\ast \mathbf{z}_t^Y} \cdot (\mathbf{R}_s^\ast \mathbf{R}_s \mathbf{h}\_\theta (\cdot) )$ (Eqs. (7)--(10) in that paper), and proposes the same cyclic dual-space integration (Algorithm 1).
The PCFM submission reproduces this formulation almost line-for-line, merely renaming the unbiased estimator as a "GSURE-based projected loss." From a technical standpoint, the novelty relative to the already accepted GTF$^2$M appears minimal.
Without explicit acknowledgment or differentiation, this raises serious novelty and dual-submission concerns for ICLR.

2. **Lack of clear differentiation.** If the authors intend PCFM as an extension of GTF$^2$M, the distinction must be mathematically or empirically meaningful. Simply replacing ENSURE with GSURE or rephrasing the projection operator is _not_ sufficient unless this change provably improves bias, variance, or convergence stability.

3. **Lack of explicit mathematical analysis for parallel MRI** In parallel MRI, directly inverting a matrix of the form $\mathbf{A}^\top \mathbf{A} + \mathbf{I}$ is typically infeasible, yet the paper provides _no_ explicit mathematical analysis of how the projection operator is approximated (e.g., number of CG iterations and effect of errors in $\mathbf{A}^+$).
Likewise, although coil sensitivity maps are only estimated in practice, there is _no_ mathematical analysis of how such estimation errors propagate through the reconstruction process or affect performance.

4. It's unclear if the sensitivity maps are estimated from under-sampled $k$-space data. It sounds that they are estimated from fully-sampled $k$-space data.

5. For an ICLR audience, the contribution should transcend the MRI domain and offer insights relevant to general inverse problems or flow-based learning under measurement constraints.
The current paper is written almost entirely as an application paper; the methodology remains MRI-specific, and the potential extension to other linear inverse problems (e.g., CT and image deblurring) is not discussed.

6. Although the authors briefly mention that the IPMI 2025 paper "focused solely on a single-coil MRI model with a feasible weighted projection operator," while the present work "introduces a rigorous GSURE-based projected CFM formulation for parallel MRI," this clarification remains insufficient.
The claimed difference is stated only at a conceptual level and lacks mathematical or experimental evidence demonstrating that the GSURE-based projection or the parallel-coil setting leads to a substantively distinct formulation, objective, or reconstruction behavior.
Without a rigorous theoretical or empirical comparison, the overlap in formulation and algorithmic structure still appears substantial.

**Questions:**

1. What is the precise relationship between PCFM and the IPMI 2025 paper GTF$^2$M? Could you clarify which parts are new or substantially modified, other than replacing ENSURE with GSURE or rephrasing the projection operator?

2. How does the GSURE-based loss differ mathematically from the ENSURE-based estimator in GTF$^2$M? Could you provide a proof or (at minimum) empirical evidence that this change improves bias, variance, or stability?

3. Is the cyclic reconstruction algorithm (forward/backward ODE with proximal data-consistency updates) identical to Algorithm 1 in GTF$^2$M, or has any component been changed (again, other than effects from replacing ENSURE with GSURE or rephrasing the projection operator)?

4. Could you provide a mathematical analysis or theoretical conditions describing how estimation errors in the sensitivity maps and approximation errors in the projection operators (e.g., due to CG iterations) affect reconstruction accuracy or stability?

5. If so, what theoretical conditions are required on the forward operator and the associated projection operators to ensure convergence or unbiasedness of the learned flow field, particularly with respect to their approximation or estimation errors?

6. Could you clarify if the sensitivity maps are estimated from under-sampled $k$-space data?

**Details Of Ethics Concerns:**

The submission appears to substantially overlap with an already accepted IPMI 2025 paper ("Unsupervised Accelerated MRI Reconstruction via Ground-Truth-Free Flow Matching"), sharing nearly identical formulations, equations, and algorithms.
The authors mention the prior work only briefly and without sufficient differentiation.
This raises potential concerns regarding duplicate or salami publication and lack of transparency about prior dissemination.

---

### Official Review · Reviewer_BTkw · 2025-10-31

**Soundness:** 3
**Presentation:** 2
**Contribution:** 2
**Rating:** 2
**Confidence:** 5

**Summary:**

The paper provides a projected CFM for parallel MRI reconstruction. The paper largely follows the single-coil derivation of Luo et al, ICML, 2025 with some modifications for multi-coil operators. The results show good performance.

**Strengths:**

- The paper derives projected CFM for multi-coil MRI setup.
- Under the experimental conditions, the method shows very good performance.

**Weaknesses:**

- The incremental contribution from Luo et al is minor. Most of the changes are related to using CG for numerically solving the projection-related formulations instead of using the closed form solutions in the single-coil setup used by Luo et al. But this does not require many major changes.
- Most of the derivations are also not properly acknowledging the previous work. As one simple example (among others) Proposition 2 and in particular eq. 13 have direct corresponding terms in GTF^2M (around eq. 9).
- The comparisons are not matched. Some methods (MoDL, ENSURE, MOI) are run with a MoDL based network, which uses a CG-based approach, while others (SSDU, Weighted SSDU, Robust SSDU, REI) use VarNet, which uses a gradient descent type optimization. It is hard to tease out if the differences are due to training or different network architectures.
- No hypothesis is provided as to why the proposed framework performs better than all other methods, in fact also beating supervised methods by a large margin. This is especially surprising given the multiple steps of numerical approximations/integrations needed in Algorithm 1.

Minor:
- The delta-function type of conditional flow considered here is not the optimal transport flow, but the one from rectified flow. OT-CFM nominally defines a 2-Wasserstein optimal transport map on q, but this paper uses the independent coupling (I-CFM) with sigma = 0, yielding the rectified flow path.
- A does not have to be rank-deficient in multi-coil MRI. Though I don't think this really affects downstream arguments, except making N(A) = {0}.

**Questions:**

From weaknesses:
- How does the performance comparisons change when different training methods use the same baseline networks?
- What is the contribution over GTF^2M?
- Why does the proposed framework so much better than all other methods, including supervised training?

---

### Official Review · Reviewer_QnzY · 2025-11-01

**Soundness:** 4
**Presentation:** 3
**Contribution:** 4
**Rating:** 8
**Confidence:** 4

**Summary:**

The paper introduces Projected Conditional Flow Matching (PCFM), a novel unsupervised generative framework for reconstructing parallel MRI data without the need for fully sampled reference images.

Building on the recently proposed Conditional Flow Matching (CFM) paradigm, the authors reformulate flow-matching losses in measurement space (k-space) by incorporating a projection operator that accounts for the non-invertibility of the MRI forward model. Using Generalized Stein’s Unbiased Risk Estimation (GSURE), they derive an unbiased unsupervised training objective that depends only on measurable quantities.

The approach is validated on parallel MRI datasets, demonstrating reconstruction quality comparable to (and sometimes exceeding) supervised diffusion-based and unrolled networks, while requiring no ground-truth images.

**Strengths:**

- **Novelty:** The paper makes a clear conceptual leap: transforming flow-matching generative modeling into an unsupervised reconstruction framework by embedding it in the measurable subspace defined by the MRI forward model.
- **Theoretical Elegance:** The derivation of Projected Conditional Flow Matching (PCFM) from Conditional Flow Matching (CFM) is mathematically rigorous and clean. The transition from supervised conditional flows → projected flows → unsupervised GSURE form (Eq. 12) is logically coherent.
- **Significance:** The ability to train high-quality MRI reconstructions entirely from undersampled data addresses one of the most pressing bottlenecks in MRI research (the scarcity of fully sampled datasets).
- **Sound experiments:** The experiments compare PCFM to several strong baselines: unrolled networks, score-based diffusion methods, and other unsupervised/self-supervised approaches, and show its superior performance. The inclusion of ablations (e.g., with/without projection, divergence correction) helps isolate which parts of the method contribute most.
- **Clarity:** The writing is dense but technically precise.

**Weaknesses:**

- The derivations, particularly around Equations 10–13, are heavy and may be opaque to readers not familiar with flow-matching or GSURE theory. The paper would benefit from a more intuitive explanation of how the GSURE term arises and what role it plays in training stability and reconstruction quality. In particular, readers may find it unclear why the unsupervised objective does not collapse to trivial zero-filled solutions, given that the network only observes undersampled k-space measurements.
- Although the experiments are solid, the datasets are relatively narrow (e.g., limited to brain and knee MRI).
- The assumption that the noise model is Gaussian may limit applicability in practical MRI settings where noise is Rician or correlated across coils.

**Questions:**

- How will PCFM behave when the sampling mask changes between training and inference?
- Since the approach is unsupervised, how exactly does the model learn a prior over realistic MR images? Is it purely from the empirical distribution of undersampled measurements, or is there any explicit regularization or implicit bias introduced by the flow architecture?
- How does this flow-matching framework compare intuitively and mathematically to diffusion-based unsupervised MRI methods?

---

### Official Review · Reviewer_5CF1 · 2025-11-05

**Soundness:** 3
**Presentation:** 3
**Contribution:** 3
**Rating:** 6
**Confidence:** 3

**Summary:**

This paper introduces an unsupervised framework for parallel MRI reconstruction called projected conditional flow matching (PCFM). The key problem it addresses is reconstructing a high-quality MR image from undersampled multi-coil k-space measurements without access to fully sampled ground truth images for training. The authors made three key contributions:
1. They formulate a projected conditional flow matching objective. This adapts the standard flow matching loss, which requires ground truth data, by projecting the error onto the range space of the forward operator's adjoint, $\mathcal{R}(A^T)$.
2. They derive an unsupervised transformation of this objective using Stein's Risk Estimator, which enables learning vector fields lead to high quality image using undersampled measurements.
3. They propose a dual-space cyclic integration algorithm for inference.

**Strengths:**

1. The proposed method is novel. The formulation of a projected flow matching objective (PCFM) to handle the rank-deficient parallel MRI operator and the subsequent "unsupervised transformation" (Proposition 2) are theoretical contributions. The derivation of the dual-space cyclic integration algorithm (Algorithm 1) for inference is also a non-trivial.
2. The evaluation is robust. It uses two distinct and standard public datasets (brain and cardiac) , compares against a wide array of recent baselines from all three categories (supervised, self-supervised, and unsupervised) , and includes ablation studies (in Appendix E) that justify the components of the proposed inference algorithm.
3. The paper demonstrates experimentally that this unsupervised method not only surpasses other unsupervised and self-supervised techniques but also performs competitively against several supervised methods on public datasets.

**Weaknesses:**

1. The method's implementation appears highly complex. It relies on numerical approximation of the Moore-Penrose pseudoinverse ($A^+$) via the Conjugate Gradient (CG) method. This approximation is used both during the unsupervised training (for the projection P=$A^+A$) and during inference (for the data consistency step). The paper states 30 CG steps are used for inference. The stability of this "inner loop" (CG) within the "outer loop" (ODE integration) and its impact on training convergence are not fully explored.
2. To me, the projection is kind of a way to recover high quality samples from undersampled measurements during the training. would it be possible to have a new baseline that is trained recovered samples?
3. It is a question if the forward operator, $A$, is changing during the training or kept fixed? Other methods dealing with degraded data keep the $A$ operator un-fixed. Would you comment on this?

**Questions:**

1. The method GTF2M in this work is better than original publication? Considering the authors of GTF2M has not released their code, would you comment on this?
2. The SSIMs for ENSURE and GTF2M at 4x acceleration are the same in Table 2. Is this a coincidence?

---

### Note · Authors · 2025-11-18

**Comment:**

We appreciate the reviewers’ thorough evaluations. PCFM is a necessary and substantive advance over GTF^2M. The theoretical reframing and the objective we optimize are fundamentally different and directly responsible for the observed performance gains.

- PCFM targets the multi‑coil (parallel) MRI problem with the operator $A = [M\,F\,S_c]_{c=1}^C$ and explicitly optimizes a projected conditional flow‑matching objective in the measurement‑visible subspace via the projector $P = A^{+}A$.
- In contrast, GTF^2M seeks an unbiased estimator of the original single‑coil CFM objective; that path is not tractable in the multi‑coil regime. PCFM’s projection changes the learning target itself rather than attempting to recover the unprojected field from undersampled measurements.
- We provide rigorous derivations connecting the dual‑space cyclic integration to the optimal solution of the projected objective (Propositions 1 and 3). This theory yields a measurement‑space marginal vector field and a closed‑form posterior step tailored to parallel MRI, and it justifies the use of a numerically approximated projection operator inside the ODE loop.
- Although the outer algorithmic scaffold resembles GTF^2M (forward–backward integration), the change in objective, target field, and projection-aware posterior fundamentally alter the learning dynamics, yielding significant performance gain of PCFM over GTF^2M on multi-coil MRI reconstruction, as already presented in the submission.

We feel that some reviewers will not appreciate our contributions. Therefore, we are withdrawing the submission.

**Withdrawal Confirmation:**

I have read and agree with the venue's withdrawal policy on behalf of myself and my co-authors.